# THE INDUCTIVE BIAS OF ReLU NETWORKS ON ORTHOGONALLY SEPARABLE DATA

**Mary Phuong & Christoph H. Lampert**
IST Austria
Am Campus 1, 3400 Klosterneuburg, Austria
`{bphuong,chl}@ist.ac.at`

We study the inductive bias of two-layer ReLU networks trained by gradient flow. We identify a class of easy-to-learn ('orthogonally separable') datasets, and characterise the solution that ReLU networks trained on such datasets converge to. Irrespective of network width, the solution turns out to be a combination of two max-margin classifiers: one corresponding to the positive data subset and one corresponding to the negative data subset.

The proof is based on the little-known concept of extremal sectors, for which we prove a number of properties in the context of orthogonal separability. In particular, we prove stationarity of activation patterns from some time $T$ onwards, which enables a reduction of the ReLU network to an ensemble of linear subnetworks.

## 1 INTRODUCTION

This paper is motivated by the problem of understanding the inductive bias of ReLU networks, or to put it plainly, understanding what it is that neural networks learn. This is a fundamental open question in neural network theory; it is also a crucial part of understanding how neural networks behave on previously unseen data (generalisation) and it could ultimately lead to rigorous a priori guarantees on neural nets' behaviour.

For a long time, the dominant way of thinking about machine learning systems was as minimisers of the empirical risk (Vapnik, 1998; Shalev-Shwartz & Ben-David, 2014). However, this paradigm has turned out to be insufficient for understanding deep learning, where many empirical risk minimisers exist, often with vastly different generalisation properties. To understand deep networks, we therefore need a more fine-grained notion of 'what the model learns'.

This has motivated the study of the implicit bias of the training procedure – the ways in which the training algorithm influences which of the empirical risk minimisers is attained. This is a productive research area, and the implicit bias has already been worked out for many linear models.[1] Notably, Soudry et al. (2018) consider a logistic regression classifier trained on linearly separable data, and show that the normalised weight vector converges to the max-margin direction. Building on their work, Ji & Telgarsky (2019a) consider deep linear networks, also trained on linearly separable data, and show that the normalised end-to-end weight vector converges to the max-margin direction. They in fact show that all first-layer neurons converge to the same 'canonical neuron' (which points in the max-margin direction). Although such impressive progress on linear models has spurred attempts at nonlinear extensions, the problem is much harder and analogous nonlinear results have been elusive.

In this work, we provide the first such inductive-bias result for ReLU networks trained on 'easy' datasets. Specifically, we

- propose orthogonal separability of datasets as a stronger form of linear separability that facilitates the study of ReLU network training,

- prove that a two-layer ReLU network trained on an orthogonally separable dataset learns a function with two distinct groups of neurons, where all neurons in each group converge to the same 'canonical neuron',

- characterise the directions of the canonical neurons, which turn out to be the max-margin directions for the positive and the negative data subset.

---

[1]A more thorough overview of related work can be found in Section 6.

The proof is based on the recently introduced concept of extremal sectors (Maennel et al., 2018) which govern the early phase of training. Our main technical contributions are a precise characterisation of extremal sectors for orthogonally separable datasets, and an invariance property which ensures that the network's activation pattern becomes fixed at some point during training. The latter allows us to treat ReLU networks late in training as ensembles of linear networks, which are much better understood. We hope that a similar proof strategy could be useful in other contexts as well.

## 2 SETTING AND ASSUMPTIONS

In this section, we introduce the learning scenario including the assumptions we make about the dataset, the model, and the training procedure. We consider binary classification. Denote the training data $\{(\mathbf{x}_i, y_i)\}_{i=1}^n$ with $\mathbf{x}_i \in \mathbb{R}^d$ and $y_i \in \{\pm 1\}$ for all $i \in [n]$. We denote by $\mathbf{X} \in \mathbb{R}^{d \times n}$ the matrix with $\{\mathbf{x}_i\}$ as columns and by $\mathbf{y} \in \mathbb{R}^n$ the vector with $\{y_i\}$ as entries.

**Orthogonally separable data.** A binary classification dataset $(\mathbf{X}, \mathbf{y})$ is called *orthogonally separable* if for all $i, j \in [n]$,

$$
\begin{aligned}
\mathbf{x}_i^\mathsf{T} \mathbf{x}_j > 0, && \text{if } y_i = y_j, \\
\mathbf{x}_i^\mathsf{T} \mathbf{x}_j \le 0, && \text{if } y_i \neq y_j.
\end{aligned}
\tag{1}
$$

In other words, a dataset is orthogonally separable iff it is linearly separable, and any training example $\mathbf{x}_i$ can serve as a linear separator. Geometrically, this means that examples with $y_i = 1$ ('positive examples') and examples with $y_i = -1$ ('negative examples') lie in opposite orthants.

**Two-layer ReLU networks.** We define two-layer width-$p$ fully-connected ReLU networks, parameterised by $\boldsymbol{\theta} \triangleq \{\mathbf{W}, \mathbf{a}\}$, as

$$
\begin{aligned}
f_{\boldsymbol{\theta}} &: \mathbb{R}^d \to \mathbb{R}, \\
f_{\boldsymbol{\theta}}(\mathbf{x}) &\triangleq \mathbf{a}^\mathsf{T} \rho(\mathbf{W}\mathbf{x}),
\end{aligned}
\tag{2}
$$

where $\mathbf{W} \triangleq [\mathbf{w}_1, \dots \mathbf{w}_p]^\mathsf{T} \in \mathbb{R}^{p \times d}$ and $\mathbf{a} \triangleq [a_1, \dots, a_p]^\mathsf{T} \in \mathbb{R}^p$ are the first- and second-layer weights of the network, and $\rho$ is the element-wise ReLU function, $\rho(\mathbf{z})_i = \max\{0, z_i\}$. We will often view the network as a collection of neurons, $\{(a_j, \mathbf{w}_j)\}_{j=1}^p$.

**Cross-entropy loss.** We assume a training loss of the form

$$
\ell(\boldsymbol{\theta}) \triangleq \sum_{i=1}^n \ell_i(f_{\boldsymbol{\theta}}(\mathbf{x}_i)), \qquad \ell_i(u) \triangleq \log(1 + \exp(-y_i u));
\tag{3}
$$

this is the standard empirical cross-entropy loss. More generally, our results hold when the loss is differentiable, $\ell_i'$ is bounded and Lipschitz continuous, and satisfies $-y_i \ell_i'(u) > 0$ for all $u \in \mathbb{R}$.

**Gradient flow training.** We assume the loss is optimised by gradient descent with infinitesimally small step size, also known as gradient flow. Under the gradient flow dynamics, the parameter trajectory is an absolutely continuous curve $\{\boldsymbol{\theta}(t) \mid t \ge 0\}$ satisfying the differential inclusion

$$
\frac{\partial \boldsymbol{\theta}(t)}{\partial t} \in -\partial \ell(\boldsymbol{\theta}(t)), \qquad \text{for almost all } t \in [0, \infty),
\tag{4}
$$

where $\partial \ell$ denotes the Clarke subdifferential (Clarke, 1975; Clarke et al., 2008) of $\ell$, an extension of the gradient to not-everywhere differentiable functions,

$$
\partial \ell(\boldsymbol{\theta}) \triangleq \text{conv}\left\{ \lim_{k \to \infty} \nabla \ell(\boldsymbol{\theta}_k) \, \middle| \, \boldsymbol{\theta}_k \to \boldsymbol{\theta} \right\}.
\tag{5}
$$

$\boldsymbol{\theta}(t)$ is the value of the parameters at time $t$, and we will use the suffix $(t)$ more generally to denote the value of some function of $\boldsymbol{\theta}$ at time $t$.

**Near-zero balanced initialisation.** We assume that the neurons $\{\mathbf{w}_j\}$ are initialised iid from the Gaussian distribution and then rescaled such that $\|\mathbf{w}_j\| \leq \lambda$, where $\lambda > 0$ is a small constant. That is, $\mathbf{w}_j = \lambda_j \mathbf{v}_j / \|\mathbf{v}_j\|$ for $\mathbf{v}_j \overset{\text{iid}}{\sim} \mathcal{N}(\mathbf{0}, \mathbf{I})$ and arbitrary $\lambda_j$ satisfying $\lambda_j \in (0, \lambda]$. We also assume that $a_j \in \{\pm\lambda_j\}$. These technical conditions ensure that the neurons are balanced and small in size, $\|\mathbf{w}_j\| = |a_j| \approx 0$, which simplifies the calculations involved in the analysis of gradient flow.

**Support examples span the full space.** We assume that the support examples of the positive data subset $\{\mathbf{x}_i \,|\, y_i = 1\}$ span the entire $\mathbb{R}^d$, and similarly that the support examples of the negative data subset $\{\mathbf{x}_i \,|\, y_i = -1\}$ span $\mathbb{R}^d$. (We formally define support examples after introducing some more notation below.)

## 3 MAIN RESULT

Under the assumptions of Section 2, the network converges to a linear combination of two max-margin neurons. Specifically, given a dataset $(\mathbf{X}, \mathbf{y})$, define the *positive and the negative max-margin vectors* $\mathbf{w}_+, \mathbf{w}_- \in \mathbb{R}^d$ as

$$\mathbf{w}_+ = \arg\min_{\mathbf{w}} \|\mathbf{w}\|^2 \quad \text{subject to} \quad \mathbf{w}^\mathsf{T}\mathbf{x}_i \geq 1 \text{ for } i : y_i = 1, \tag{6}$$

$$\mathbf{w}_- = \arg\min_{\mathbf{w}} \|\mathbf{w}\|^2 \quad \text{subject to} \quad \mathbf{w}^\mathsf{T}\mathbf{x}_i \geq 1 \text{ for } i : y_i = -1. \tag{7}$$

We call examples which attain equality in eqs. (6) and (7) *positive support examples* and *negative support examples* respectively. We now state the main result.

**Theorem 1.** *Let $f_{\boldsymbol{\theta}}$ be a two-layer width-$p$ ReLU network trained by gradient flow with the cross-entropy loss, initialised near-zero and balanced. Consider an orthogonally separable dataset $(\mathbf{X}, \mathbf{y})$ such that its positive support examples span $\mathbb{R}^d$, and its negative support examples also span $\mathbb{R}^d$. For almost all such datasets[2] and with probability $1 - 1/2^p$ over the random initialisation,*

$$\left\| \frac{\mathbf{W}(t)}{\|\mathbf{W}(t)\|_F} - \left(\mathbf{u}\mathbf{w}_+^\mathsf{T} + \mathbf{z}\mathbf{w}_-^\mathsf{T}\right) \right\|_F \to 0, \qquad \text{as } t \to \infty, \tag{8}$$

*for some $\mathbf{u}, \mathbf{z} \in \mathbb{R}_+^p$ such that either $u_i = 0$ or $z_i = 0$ for all $i \in [p]$. Also,*

$$\left\| \frac{\mathbf{a}(t)}{\|\mathbf{a}(t)\|} - (\mathbf{u}\|\mathbf{w}_+\| - \mathbf{z}\|\mathbf{w}_-\|) \right\| \to 0, \qquad \text{as } t \to \infty. \tag{9}$$

The theorem says that each neuron (row of $\mathbf{W}$), properly normalised, converges either to a scalar multiple of the positive max-margin direction, $u_i\mathbf{w}_+$, or to a scalar multiple of the negative max-margin direction, $z_i\mathbf{w}_-$. In other words, there are asymptotically only two distinct types of neurons, and the network could in principle be pruned down to a width of just two. These two 'canonical neurons' moreover have an explicit characterisation, given by eqs. (6) and (7).

As for the second-layer weights, the magnitude of each $a_j$ equals the norm of the respective $\mathbf{w}_j$, and the sign of $a_j$ is $+1$ if $\mathbf{w}_j$ approaches $\mathbf{w}_+$ and $-1$ if $\mathbf{w}_j$ approaches $\mathbf{w}_-$.

The following corollary summarises the above in terms of the function learnt by the network.

**Corollary 1.** *Under the conditions of Theorem 1, there exist constants $u, z \geq 0$ such that*

$$\frac{f_{\boldsymbol{\theta}(t)}(\mathbf{x})}{\|\boldsymbol{\theta}(t)\|^2} \to u\rho(\mathbf{w}_+^\mathsf{T}\mathbf{x}) - z\rho(\mathbf{w}_-^\mathsf{T}\mathbf{x}), \qquad \text{as } t \to \infty. \tag{10}$$

### 3.1 DISCUSSION OF ASSUMPTIONS

Many of our assumptions are technical, serving to simplify the analysis while detracting little from the result's relevance[3]. These include infinitesimal step size (gradient flow), balancedness at initialisation and the condition on support span. The first two could potentially be relaxed to their

---

[2]Formally, this means that if $\{\mathbf{x}_i\}$ are sampled from any distribution with a density wrt. the Lebesgue measure, then the theorem (treated as an implication) holds with probability one wrt. the data.

[3]We verify experimentally in Section 5.1 that these assumptions are indeed not crucial.

approximate counterparts, i.e. gradient descent with a small constant step size and approximate balancedness (Arora et al., 2019). The assumption that support vectors span $\mathbb{R}^d$ comes from Ji & Telgarsky (2019a); Soudry et al. (2018). It seems to us that it could be lifted, though we have not investigated this possibility in depth.

Two assumptions that deserve more attention are near-zero initialisation and orthogonal separability; both are crucial for the result to hold. Near-zero initialisation grants neurons high directional mobility early in training, allowing them to cluster close to the canonical directions. Orthogonal separability ensures that the canonical directions are 'easy to find' by local descent. In prior work, which considered linear networks, this role is fulfilled by linear separability. The reason we need a stronger condition is that ReLU updates are more local compared to linear updates: a linear neuron takes into account all examples in the training set, whereas a ReLU neuron updates only on examples in its positive half-plane (its active examples). ReLU neurons therefore easily get stuck in a variety of directions, unless the data is highly structured.

## 4 Proof sketch

In the analysis, we distinguish between two phases of training. The first phase takes place close to the origin, $\|\boldsymbol{\theta}\| \approx 0$. In this phase, while neurons move little in the absolute sense, they converge in direction to certain regions of the weight space called *extremal sectors*.

### 4.1 Convergence to extremal sectors

(All definitions and results in this subsection are by Maennel et al. (2018). We need them later on.)

Sectors are regions in weight space corresponding to different activation patterns. They are important for understanding neuron dynamics: roughly speaking, neurons in the same sector move in the same direction.

**Definition 1** (Sector). *The* sector *associated to a sequence of signs* $\boldsymbol{\sigma} \in \{-1, 0, 1\}^n$ *is the region in input space defined as*

$$\mathcal{S}_{\boldsymbol{\sigma}} \triangleq \{\mathbf{w} \in \mathbb{R}^d \mid \operatorname{sign} \mathbf{w}^\mathsf{T} \mathbf{x}_i = \sigma_i,\ i \in [n]\}. \tag{11}$$

*We may also refer to the sign sequence* $\boldsymbol{\sigma}$ *itself as a sector.*

Some sectors are attractors early in training, i.e. neurons tend to converge to them. Such attracting sectors are called *extremal sectors*. To give a formal definition, we first introduce the function $G : \mathbb{S}^{d-1} \to \mathbb{R}$,

$$G(\mathbf{w}) \triangleq -\sum_{i=1}^{n} \ell_i'(0) \cdot \rho(\mathbf{w}^\mathsf{T} \mathbf{x}_i). \tag{12}$$

Intuitively, (normalised) neurons early in training behave as if they were locally optimising $G$, they therefore tend to cluster around the local optima of $G$. We formally define extremal sectors as sectors containing these local optima.

**Definition 2** (Extremal directions and sectors). *We say that* $\mathbf{w} \in \mathbb{S}^{d-1}$ *is a* positive extremal direction, *if it is a strict local maximum of* $G$. *We say that* $\mathbf{w}$ *is a* negative extremal direction *if it is a strict local minimum of* $G$. *A sector is called (positive/negative)* extremal, *if it contains a (positive/negative) extremal direction.*

The following lemma (Maennel et al., 2018, Lemma 5) shows that all neurons either turn off, i.e. become deactivated for all training examples and stop updating, or converge to extremal sectors.

**Lemma 1.** *Let a two-layer ReLU network* $f_{\boldsymbol{\theta}}$ *be balanced at initialisation and trained by gradient flow. Assume that the loss derivative* $\ell_i'$ *is Lipschitz continuous. Then, for almost all datasets and almost all initialisations with* $\lambda$ *small enough, there exists a time* $T$ *such that each neuron satisfies one of these three conditions:*

- $\mathbf{w}_j(T) \in \mathcal{S}_{\boldsymbol{\sigma}}$ *where* $\boldsymbol{\sigma} \leq \mathbf{0}$ *and so* $\mathbf{w}_j$ *remains constant for* $t \geq T$, *or*

- $a_j(T) > 0$ *and* $\mathbf{w}_j(T) \in \mathcal{S}_{\boldsymbol{\sigma}}$ *where* $\boldsymbol{\sigma}$ *is a positive extremal sector, or*

- $a_j(T) < 0$ *and* $\mathbf{w}_j(T) \in \mathcal{S}_{\boldsymbol{\sigma}}$ *where* $\boldsymbol{\sigma}$ *is a negative extremal sector.*

## 4.2 Orthogonal separability: Two absorbing extremal sectors

Lemma 1 shows that by the end of the early phase of training, neurons have converged to extremal sectors. Although eq. (12) shows that the number of extremal sectors depends only on the data (i.e. is independent of model expressivity), it is a priori unclear how many extremal sectors there are for a given dataset, or what happens once neurons have converged to extremal sectors. We now answer both of these questions for orthogonally separable datasets.

First, we claim that for orthogonally separable datasets, there are only two extremal sectors, one corresponding to the positive data subset and one corresponding to the negative data subset. That is, by converging to an extremal sector, neurons 'choose' whether to activate for positive examples or for negative examples. They thus naturally form two groups of similar neurons.

**Lemma 2.** *In the setting of Theorem 1, there is exactly one positive extremal direction and exactly one negative extremal direction. The positive extremal sector $\boldsymbol{\sigma}^+$ is given by*

$$\sigma_j^+ = \begin{cases} 1, & \text{if } y_j = 1, \\ -1, & \text{if } y_j = -1 \text{ and } \mathbf{x}_j^\mathsf{T} \mathbf{x}_i < 0 \text{ for some } i \text{ with } y_i = 1, \\ 0, & \text{if } y_j = -1 \text{ and } \mathbf{x}_j^\mathsf{T} \mathbf{x}_i = 0 \text{ for all } i \text{ with } y_i = 1, \end{cases} \tag{13}$$

*and the negative extremal sector $\boldsymbol{\sigma}^-$ is given by*

$$\sigma_j^- = \begin{cases} 1, & \text{if } y_j = -1, \\ -1, & \text{if } y_j = 1 \text{ and } \mathbf{x}_j^\mathsf{T} \mathbf{x}_i < 0 \text{ for some } i \text{ with } y_i = -1, \\ 0, & \text{if } y_j = 1 \text{ and } \mathbf{x}_j^\mathsf{T} \mathbf{x}_i = 0 \text{ for all } i \text{ with } y_i = -1. \end{cases} \tag{14}$$

Second, we show that once a neuron reaches an extremal sector, it remains in the sector forever, i.e. its activation pattern remains fixed for the rest of training.

**Lemma 3.** *Assume the setting of Theorem 1. If at time $T$ the neuron $(a_j, \mathbf{w}_j)$ satisfies $a_j(T) > 0$ and $\mathbf{w}_j(T) \in \mathcal{S}_{\boldsymbol{\sigma}}$, where $\boldsymbol{\sigma}$ is the positive extremal sector (eq. (13)), then for $t \geq T$, $\mathbf{w}_j(t) \in \mathcal{S}_{\boldsymbol{\sigma}}$. The same holds if $a_j(T) < 0$ and $\boldsymbol{\sigma}$ is the negative extremal sector (eq. (14)).*

## 4.3 Proof of Theorem 1

Once neurons enter their respective absorbing sectors, the second phase of training begins. In this phase, the network's activation patterns are fixed: some neurons are active and update on the positive examples, while the others are active and update on the negative examples. The network thus behaves like an ensemble of independent linear subnetworks trained on subsets of the data. Once this happens, it becomes possible to apply existing results for linear networks; in particular, each subnetwork converges to its respective max-margin classifier.

We give more details in the proof below.

*Proof of Theorem 1.* By Lemmas 1 and 2, there exists a time $T$ such that each neuron satisfies either

- $\mathbf{w}_j(T) \in \mathcal{S}_{\boldsymbol{\sigma}}$ where $\boldsymbol{\sigma} \leq \mathbf{0}$ and $\mathbf{w}_j$ remains constant for $t \geq T$, or

- $a_j(T) > 0$ and $\mathbf{w}_j(T) \in \mathcal{S}_{\boldsymbol{\sigma}^+}$, or

- $a_j(T) < 0$ and $\mathbf{w}_j(T) \in \mathcal{S}_{\boldsymbol{\sigma}^-}$,

where $\boldsymbol{\sigma}^+, \boldsymbol{\sigma}^-$ are the unique positive and negative extremal sectors given by eqs. (13) and (14). Denote by $\mathcal{J}_0, \mathcal{J}_+, \mathcal{J}_-$ the sets of neurons satisfying the first, the second, and the third condition respectively. By Lemma 3, if $j \in \mathcal{J}_+$ then $\mathbf{w}_j(t) \in \mathcal{S}_{\boldsymbol{\sigma}^+}$ for all $t \geq T$, and if $j \in \mathcal{J}_-$ then $\mathbf{w}_j(t) \in \mathcal{S}_{\boldsymbol{\sigma}^-}$ for $t \geq T$. Hence, for $t \geq T$, if $\mathbf{x}_i$ is such that $y_i = 1$ then

$$f_{\boldsymbol{\theta}}(\mathbf{x}_i) \triangleq \sum_{j \in [p]} a_j \rho(\mathbf{w}_j^\mathsf{T} \mathbf{x}_i) = \sum_{j \in \mathcal{J}_+} a_j \mathbf{w}_j^\mathsf{T} \mathbf{x}_i. \tag{15}$$

Combined with Lemma A.3, this implies that for $k \in \mathcal{J}_+$,

$$
\begin{aligned}
\frac{\partial a_k}{\partial t} &= -\sum_{i:y_i=1} \ell_i'\left(\sum_{j\in\mathcal{J}_+} a_j \mathbf{w}_j^\mathsf{T} \mathbf{x}_i\right) \cdot \mathbf{w}_k^\mathsf{T} \mathbf{x}_i, \\
\frac{\partial \mathbf{w}_k}{\partial t} &= -\sum_{i:y_i=1} \ell_i'\left(\sum_{j\in\mathcal{J}_+} a_j \mathbf{w}_j^\mathsf{T} \mathbf{x}_i\right) \cdot a_k \mathbf{x}_i,
\end{aligned}
\tag{16}
$$

(where we have used that $\mathbf{P_w x}_i = \mathbf{x}_i$ for $i$ with $y_i = 1$ due to positive extremality). From eq. (16) it follows that the evolution of neurons in $\mathcal{J}_+$ depends only on positive examples and other neurons in $\mathcal{J}_+$. The neurons behave linearly on the positive data subset, while ignoring the negative subset. The same argument shows that the evolution of neurons in $\mathcal{J}_-$ depends only on other neurons in $\mathcal{J}_-$ and the negative data subset, on which the neurons act linearly. In other words, from time $T$ onwards the ReLU network decomposes into a constant part and two independent linear networks, one trained on the positive data subset and the other trained on the negative data subset.

We can therefore apply existing max-margin convergence results for linear networks to each of the linear subnetworks. Denote by $\mathbf{W}^\mathsf{T} = [\mathbf{W}_0^\mathsf{T}, \mathbf{W}_+^\mathsf{T}, \mathbf{W}_-^\mathsf{T}]$ the three parts of the weight matrix. Then by (Ji & Telgarsky, 2019a, Theorems 2.2 and 2.8) and (Ji & Telgarsky, 2020, Theorem 3.1), there exist vectors $\bar{\mathbf{u}}, \bar{\mathbf{z}}$, such that

$$
\left\| \frac{\mathbf{W}_+(t)}{\|\mathbf{W}_+(t)\|_F} - \bar{\mathbf{u}} \mathbf{w}_+^\mathsf{T} \right\|_F \to 0, \qquad \text{as } t \to \infty, \tag{17}
$$

$$
\left\| \frac{\mathbf{W}_-(t)}{\|\mathbf{W}_-(t)\|_F} - \bar{\mathbf{z}} \mathbf{w}_-^\mathsf{T} \right\|_F \to 0, \qquad \text{as } t \to \infty. \tag{18}
$$

(We allow $\bar{\mathbf{u}}, \bar{\mathbf{z}} \in \mathbb{R}^0$ to account for the fact that $\mathcal{J}_+, \mathcal{J}_-$ may be empty). We now need to relate $\|\mathbf{W}_+\|_F$ and $\|\mathbf{W}_-\|_F$ to $\|\mathbf{W}\|_F$. In particular, it will be useful to show that $\|\mathbf{W}_+(t)\|_F^2 / \log t$ has a limit as $t \to \infty$; the same is true for $\|\mathbf{W}_-(t)\|_F^2 / \log t$ (by the same argument). If $\mathcal{J}_+$ or $\mathcal{J}_-$ is empty, this is trivially true and the limit is 0. Otherwise, consider the learning of the positive linear subnetwork, whose objective is effectively $\ell^+(\boldsymbol{\theta}) := \sum_{i:y_i=1} \ell_i(f_{\boldsymbol{\theta}}(\mathbf{x}_i))$. By (Ji & Telgarsky, 2019a, Theorem 2.2), we know that $\ell^+(\boldsymbol{\theta}(t)) \to 0$ as $t \to \infty$. Following (Lyu & Li, 2020, Definition A.3), define

$$
\tilde{\gamma}(\boldsymbol{\theta}) \triangleq \frac{g(\log 1/\ell^+(\boldsymbol{\theta}))}{2\|\mathbf{W}_+\|_F^2}, \tag{19}
$$

where $g(q) := -\log\left(\exp(\exp(-q)) - 1\right)$ for the cross-entropy loss. Then

$$
\frac{\|\mathbf{W}_+(t)\|_F^2}{\log t} = \frac{g(\log 1/\ell^+(\boldsymbol{\theta}(t)))}{2\tilde{\gamma}(t)\log t} = \frac{-\log\left(\exp(\ell^+(\boldsymbol{\theta}(t))) - 1\right)}{2\tilde{\gamma}(t)\log t}. \tag{20}
$$

Using the Taylor expansion $\exp(u) = 1 + \Theta(u)$ for $u \to 0$ and (Lyu & Li, 2020, Corollary A.11), we obtain

$$
\frac{\|\mathbf{W}_+(t)\|_F^2}{\log t} = \frac{-\log\Theta(\ell^+(\boldsymbol{\theta}(t)))}{2\tilde{\gamma}(t)\log t} = \frac{\log\Theta(t\log t)}{2\tilde{\gamma}(t)\log t} = \frac{1}{2\tilde{\gamma}(t)}\left(\frac{\Theta(1) + \log\log t}{\log t} + 1\right). \tag{21}
$$

By (Lyu & Li, 2020, Theorem A.7:1), $\tilde{\gamma}$ is increasing in $t$ and hence converges; it follows that $\|\mathbf{W}_+(t)\|_F^2 / \log t$ has a limit. By (Lyu & Li, 2020, Corollary A.11), $\|\mathbf{W}_+(t)\|_F^2 = \Theta(\log t)$, implying that the limit is finite and strictly positive. We will denote it by $\nu_+$ and the analogous quantity for $\mathbf{W}_-$ by $\nu_-$.

We now return to the main thread of the proof. We analyse the convergence of $\mathbf{W}(t)/\|\mathbf{W}(t)\|_F$ by analysing $\mathbf{W}_0/\|\mathbf{W}(t)\|_F$, $\mathbf{W}_+(t)/\|\mathbf{W}(t)\|_F$ and $\mathbf{W}_-(t)/\|\mathbf{W}(t)\|_F$ in turn. Since $\|\mathbf{W}(t)\|_F^2 = \|\mathbf{W}_0\|_F^2 + \|\mathbf{W}_+(t)\|_F^2 + \|\mathbf{W}_-(t)\|_F^2$,

$$
\lim_{t\to\infty} \frac{\|\mathbf{W}(t)\|_F^2}{\log t} = \nu_+ + \nu_-. \tag{22}
$$

Now observe that with probability at least $1 - 1/2^p$ over the random initialisation, $\nu_+ + \nu_- > 0$ (or equivalently, $\mathcal{J}_+ \cup \mathcal{J}_- \neq \emptyset$). To prove this, let $\mathbf{x}_{i+}$ be any training example with $y_{i+} = 1$ and let $\mathbf{x}_{i-}$

be any training example with $y_{i-} = -1$. Then by Lemma B.1, if a neuron $(a_j, \mathbf{w}_j)$ is initialised such that $a_j(0) > 0$ and $\mathbf{w}_j(0)^\mathsf{T} \mathbf{x}_{i+} > 0$ then for $t \geq 0$, $\mathbf{w}_j(t)^\mathsf{T} \mathbf{x}_{i+} > 0$. This holds in particular at time $T$. The neuron $j$ thus cannot be in $\mathcal{J}_0$ nor $\mathcal{J}_-$, implying $j \in \mathcal{J}_+$. Similarly, if the neuron is initialised such that $a_j(0) < 0$ and $\mathbf{w}_j(0)^\mathsf{T} \mathbf{x}_{i-} > 0$, then $j \in \mathcal{J}_-$. The probability that one of the two initialisations occurs for a single neuron $j$ is $1/2$, as $\mathbb{P}_{\mathbf{w}_j}\left[\mathbf{w}_j(0)^\mathsf{T} \mathbf{x} > 0\right] = 1/2$ for any fixed $\mathbf{x}$. Hence, the probability that $j \in \mathcal{J}_0$ is at most $1 - 1/2 = 1/2$, and the probability that $[p] \subseteq \mathcal{J}_0$ is at most $1/2^p$.

It follows that with probability at least $1 - 1/2^p$,

$$\frac{\mathbf{W}_0}{\|\mathbf{W}(t)\|_F} \to \mathbf{0}, \qquad \text{as } t \to \infty. \tag{23}$$

Also, by eqs. (17) and (22),

$$\frac{\mathbf{W}_+(t)}{\|\mathbf{W}(t)\|_F} = \frac{\mathbf{W}_+(t)}{\|\mathbf{W}_+(t)\|_F} \cdot \frac{\|\mathbf{W}_+(t)\|_F / \sqrt{\log t}}{\|\mathbf{W}(t)\|_F / \sqrt{\log t}} \to \frac{\sqrt{\nu_+}}{\sqrt{\nu_+ + \nu_-}} \bar{\mathbf{u}} \mathbf{w}_+^\mathsf{T}, \tag{24}$$

and similarly

$$\frac{\mathbf{W}_-(t)}{\|\mathbf{W}(t)\|_F} \to \frac{\sqrt{\nu_-}}{\sqrt{\nu_+ + \nu_-}} \bar{\mathbf{z}} \mathbf{w}_-^\mathsf{T}. \tag{25}$$

For $j \in \mathcal{J}_+$ and $t \geq T$ we moreover know that if $y_i = 1$ then $\mathbf{w}_j(t)^\mathsf{T} \mathbf{x}_i > 0$ because $\mathbf{w}_j(t) \in \mathcal{S}_{\boldsymbol{\sigma}+}$. As the same property holds for $\mathbf{w}_+$, it follows that $\bar{u}_j \geq 0$. By a similar argument, $\bar{z}_j \geq 0$. Combining the last three equations then proves eq. (8).

As for eq. (9), we know by Lemma A.4 that $a_j(t) = s_j \|\mathbf{w}_j(t)\|$ for some $s_j \in \{\pm 1\}$, implying $\|\mathbf{a}(t)\| = \|\mathbf{W}(t)\|_F$. Hence, for $j \in \mathcal{J}_+$,

$$\frac{a_j(t)}{\|\mathbf{a}(t)\|} = \frac{s_j \|\mathbf{w}_j(t)\|}{\|\mathbf{W}(t)\|_F} \to s_j \|u_j \mathbf{w}_+^\mathsf{T}\| \tag{26}$$

by eq. (24), where $u_j \geq 0$. For $j \in \mathcal{J}_+$ we also know that $a_j(t) \geq 0$, so $s_j = 1$ and

$$\frac{a_j(t)}{\|\mathbf{a}(t)\|} \to u_j \|\mathbf{w}_+\|. \tag{27}$$

By a similar argument, we obtain that for $j \in \mathcal{J}_-$,

$$\frac{a_j(t)}{\|\mathbf{a}(t)\|} \to -z_j \|\mathbf{w}_-\|. \tag{28}$$

Finally, for $j \in \mathcal{J}_0$, $a_j(t)$ is constant and so

$$\frac{a_j(t)}{\|\mathbf{a}(t)\|} \to 0. \tag{29}$$

$\square$

## 5 EXPERIMENTS

In this section, we first verify that the theoretical result (Theorem 1) is predictive of experimental outcomes, even when some technical assumptions are violated. Second, we present evidence that a similar result may hold for deeper networks as well, although this goes beyond Theorem 1.

### 5.1 TWO-LAYER NETWORKS

To see how well the theory holds up, we train a two-layer ReLU network with 100 neurons on a synthetic orthogonally separable dataset consisting of 500 examples in $\mathbb{R}^{20}$. The dataset is constructed from an iid Gaussian dataset by filtering, to ensure orthogonal separability and $\mathbf{w}_+ \not\approx -\mathbf{w}_-$ (for visualisation purposes). Specifically, let $\mathbf{z} := [1, -1, \ldots, 1, -1]$. A Gaussian-sampled point $\mathbf{x}$ is included with label $+1$ if it lies in the first orthant and $\mathbf{x}^\mathsf{T} \mathbf{z} \geq 0$, included with label $-1$ if it lies in the orthant opposite to the first and $\mathbf{x}^\mathsf{T} \mathbf{z} \geq 0$, and discarded otherwise.

We train by stochastic gradient descent with batch size 50 and a learning rate of $0.1$ for 500 epochs. At initialisation, we multiply all weights by $0.05$. This reflects a setting where both key assumptions of Theorem 1 – orthogonal separability and small initialisation – hold, while the other assumptions are relaxed to approach real-life practice.

Figure 1 shows the results. Figure 1a shows the top 10 singular values of the first-layer weight matrix $\mathbf{W} \in \mathbb{R}^{100 \times 20}$ after training. We see that despite its size, the matrix has rank only two: all singular values except the first two are effectively zero. This is exactly as predicted by the theorem. Furthermore, when we project the neurons on the positive-variance dimensions (Figure 1b), we see that they align along two main directions. To see how well these directions align with the predicted max-margin directions, we compute the correlation (normalised inner product) of each neuron with its respective max-margin direction. Figure 1c shows the histogram of these correlations. We see that the correlation is generally high, above 0.9 for most neurons. Overall we find very good agreement with theory.

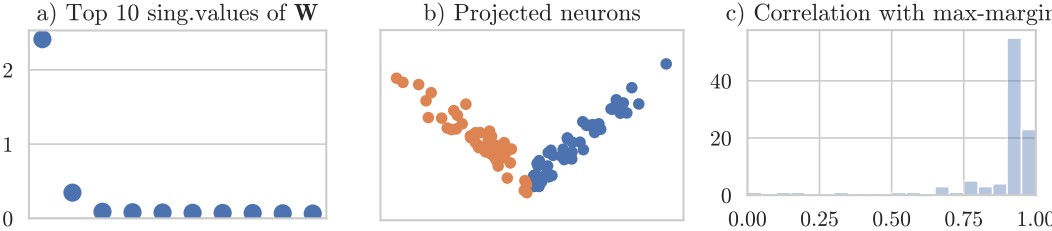

Figure 1: **a)** The 10 largest singular values of the first-layer weight matrix $\mathbf{W}$ after training. Each dot represents one singular value. **b)** Neurons (rows of $\mathbf{W}$) projected on the top two singular dimensions. Orange (or blue) dots represent neurons with $a_j > 0$ (or $a_j < 0$). **c)** Histogram of correlations between each neuron and its respective max-margin direction. (There are 100 neurons in total).

## 5.2 DEEPER NETWORKS

We now explore the behaviour of deeper networks on orthogonally separable data. We train a residual network rather than a fully-connected one. The reason for this is that fully-connected networks with small initialisation are hard to train: early in training, the gradients are vanishingly small but then grow very quickly. We therefore found setting a numerically stable learning rate rather delicate.

We consider a residual network $f_{\boldsymbol{\theta}} : \mathbb{R}^d \to \mathbb{R}$ parameterised by $\boldsymbol{\theta} \triangleq \{\mathbf{W}_1, \ldots, \mathbf{W}_L\}$, of the form

$$
\begin{aligned}
f_{\boldsymbol{\theta}}^1(\mathbf{x}) &= \mathbf{W}_1 \mathbf{x}, \\
f_{\boldsymbol{\theta}}^l(\mathbf{x}) &= f_{\boldsymbol{\theta}}^{l-1}(\mathbf{x}) + \mathbf{W}_l \rho(f_{\boldsymbol{\theta}}^{l-1}(\mathbf{x})), \qquad \text{for } l \in [2, L-1], \\
f_{\boldsymbol{\theta}}(\mathbf{x}) &= \mathbf{W}_L \rho(f_{\boldsymbol{\theta}}^{L-1}(\mathbf{x})),
\end{aligned} \tag{30}
$$

where $p$ is the network's width, and $\mathbf{W}_1 \in \mathbb{R}^{p \times d}$, $\mathbf{W}_l \in \mathbb{R}^{p \times p}$ and $\mathbf{W}_L \in \mathbb{R}^{1 \times p}$ are its weights.

We train such a four-layer residual net with width 100 on the same dataset and using the same optimiser and hyper-parameters as in Section 5.1. Figure 2 shows the results. The results are very similar to what we observe for two-layer nets: the weight matrices are all rank two (Figure 2a-c), and the weight matrices' rows align in two main directions (Figure 2d-f). It is unclear what these directions are for the intermediate layers of the network, but for the first layer, we conjecture it is again the max-margin directions, as suggested by Figure 2g.

## 6 RELATED WORK

There is a lot of prior work on the implicit bias of gradient descent for various linear models. For logistic regression, Soudry et al. (2018) show that assuming an exponentially-tailed loss and linearly separable data, the normalised weight vector converges to the max-margin direction. Ji & Telgarsky (2019b) extend this result to non-separable data, Nacson et al. (2019) extend it to super-polynomially-tailed losses, and Gunasekar et al. (2018a) considers different optimisation algorithms. For deep linear networks, Ji & Telgarsky (2019a) show that the end-to-end weight matrix converges

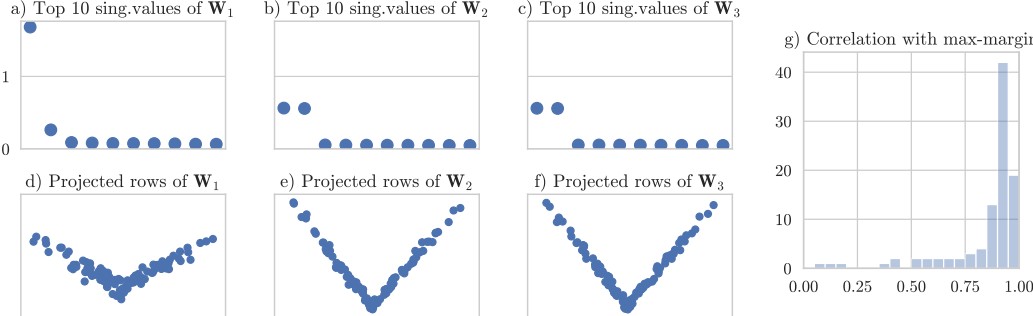

Figure 2: **a-c)** The 10 largest singular values of the first-, second- and third-layer weight matrix $\mathbf{W}_l$ after training. Each dot represents one singular value. **d-f)** Neurons (rows of $\mathbf{W}_l$) projected on the respective top two singular dimensions. **g)** Histogram of correlations between each first-layer neuron and the closest max-margin direction. (There are 100 neurons in total).

to the max-margin solution and consecutive weight matrices align. Gunasekar et al. (2018b) consider linear convolutional nets and prove convergence to a predictor related to the $\ell_{2/L}$ bridge penalty.

A few papers have started addressing the implicit bias problem for nonlinear (homogeneous or ReLU) networks. The problem is much harder and hence requires stronger assumptions. Lyu & Li (2020) and Ji & Telgarsky (2020) assume that at some point during training, the network attains perfect classification accuracy. Training from this point onward, Ji & Telgarsky (2020) show that the network parameters converge in direction. Lyu & Li (2020) show that this direction is a critical KKT point of the (nonlinear) max-margin problem. A complementary approach is taken by Maennel et al. (2018) who analyse the very early phase of training, when the weights are close to the origin. For two-layer networks, they show convergence of neurons to extremal sectors. Our work can be seen as a first step towards bridging the very early and the very late phase of training.

Zooming out a bit, there is also work motivated by similar questions, but taking a different approach. For example, Li & Liang (2018) show that two-layer ReLU nets trained on structured data converge to a solution that generalises well. Like ours, their analysis requires that the network's activation patterns change little, but they achieve it by containing training in the neighbourhood of the (relatively large) initialisation (this is the standard lazy training argument Chizat et al. (2019)). In contrast, we initialise much closer to zero, allowing the neurons to move more. Another related paper is Chizat & Bach (2020). Using a mean-field analysis, the authors show that infinite-width two-layer ReLU nets converge to max-margin classifiers in a certain non-Hilbertian function space.

## 7 CONCLUSION

In this work, we prove that two-layer ReLU nets trained by gradient flow on orthogonally separable data converge to a combination of the positive and the negative max-margin classifier. To our knowledge, this is the first result characterising the inductive bias of training neural networks with ReLU nonlinearities, that does not require infinite width or huge overparameterisation.

The proof rests on a distinction between two phases of learning: an early phase, in which neurons specialise, and a late phase, in which the network's activation pattern is fixed and hence it behaves like an ensemble of linear subnetworks. This approach enables us to understand nonlinear ReLU networks in terms of the much better understood linear networks. Our hope is that a similar strategy will prove fruitful in the context of deeper networks and more complicated datasets as well.

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

## A  BASIC LEMMAS

This section collects a few lemmas useful for proofs. We assume the same setting and notation as Sections 2 and 4. In addition, we denote by $\mathbf{P_w}$ the orthogonal projection onto $\text{span}\{\mathbf{x}_i \mid \mathbf{w}^\mathsf{T}\mathbf{x}_i = 0\}^\perp$, and by $g : \mathbb{R}^d \to \mathbb{R}^d$,

$$g(\mathbf{w}) \triangleq -\sum_{i=1}^n \ell_i'(0) \cdot \mathbb{1}\{\mathbf{w}^\mathsf{T}\mathbf{x}_i > 0\} \, \mathbf{P_w}\mathbf{x}_i. \tag{31}$$

### A.1  LEMMAS ABOUT SECTORS

The following lemma gives a necessary condition for a vector to be an extremal direction.

**Lemma A.1.** *If $\mathbf{w} \in \mathbb{S}^{d-1}$ is an extremal direction, then $g(\mathbf{w}) = C\mathbf{w}$ for some constant $C$.*

*Proof.* Let $\hat{\mathbf{w}} \in \mathbb{S}^{d-1}$ be a positive extremal direction (the negative case is analogous), and let $\hat{\mathbf{w}} \in \mathcal{S}_{\hat{\boldsymbol{\sigma}}}$. A sector is called *open* if $\sigma_i \neq 0$ for all $i \in [n]$. Denote by $\mathcal{A}(\hat{\boldsymbol{\sigma}})$ the set of all open sectors adjacent to $\hat{\boldsymbol{\sigma}}$,

$$\mathcal{A}(\hat{\boldsymbol{\sigma}}) := \left\{ \boldsymbol{\sigma} \in \{\pm 1\}^n \,\middle|\, \max_{i \in [n]} |\sigma_i - \hat{\sigma}_i| \leq 1 \right\}. \tag{32}$$

Since $\hat{\mathbf{w}}$ is a local maximum of $G$ and $G$ is sector-wise linear, $\hat{\mathbf{w}}$ maximises $G$ when constrained to (the closure of) any adjacent sector, i.e. for any $\boldsymbol{\sigma} \in \mathcal{A}(\hat{\boldsymbol{\sigma}})$,

$$\hat{\mathbf{w}} = \arg\max_{\mathbf{w}} G(\mathbf{w}), \text{ subject to } \|\mathbf{w}\|^2 = 1,$$
$$\sigma_i \mathbf{w}^\mathsf{T}\mathbf{x}_i \geq 0 \text{ for all } i \in [n]. \tag{33}$$

For $\mathbf{w}$ in the feasible region, $G$ can be treated as a linear function with $\nabla G(\mathbf{w}) = g(\mathbf{w}_{\boldsymbol{\sigma}})$ where $\mathbf{w}_{\boldsymbol{\sigma}}$ is any vector such that $\mathbf{w}_{\boldsymbol{\sigma}} \in \mathcal{S}_{\boldsymbol{\sigma}}$. Hence, the necessary first-order KKT conditions for the problem (33) are

$$g(\mathbf{w}_{\boldsymbol{\sigma}}) = C\hat{\mathbf{w}} - \sum_{i=1}^n \lambda_i \sigma_i \mathbf{x}_i, \tag{34}$$

where $\lambda_i \geq 0$ for all $i$, but $\lambda_i \neq 0$ requires that the corresponding constraint is tight, $\sigma_i \hat{\mathbf{w}}^\mathsf{T}\mathbf{x}_i = 0$. It follows that $\mathbf{P}_{\hat{\mathbf{w}}} \lambda_i \sigma_i \mathbf{x}_i = \mathbf{0}$. Multiplying eq. (34) from the left by $\mathbf{P}_{\hat{\mathbf{w}}}$ therefore yields

$$C\hat{\mathbf{w}} = \mathbf{P}_{\hat{\mathbf{w}}}\, g(\mathbf{w}_{\boldsymbol{\sigma}}) = -\sum_{i=1}^n \ell_i'(0) \cdot \mathbb{1}\{\sigma_i = 1\}\, \mathbf{P}_{\hat{\mathbf{w}}}\mathbf{x}_i. \tag{35}$$

By adjacency, $\sigma_i = \hat{\sigma}_i$ whenever $\hat{\sigma}_i \in \{\pm 1\}$, so they can differ only when $\hat{\sigma}_i = 0$, i.e. when $\mathbf{P}_{\hat{\mathbf{w}}}\mathbf{x}_i = \mathbf{0}$. It follows that

$$C\hat{\mathbf{w}} = -\sum_{i=1}^n \ell_i'(0) \cdot \mathbb{1}\{\hat{\sigma}_i = 1\}\, \mathbf{P}_{\hat{\mathbf{w}}}\mathbf{x}_i = g(\hat{\mathbf{w}}). \tag{36}$$

$\square$

The following lemma describes the local behaviour of the function $G$ (defined in eq. (12)).

**Lemma A.2.** *For $\mathbf{w} \in \mathbb{S}^{d-1}$ and $\mathbf{v} \in \mathbb{R}^d$, there exists $\epsilon_{\max} > 0$ such that for $\epsilon \in [0, \epsilon_{\max}]$,*

$$G\left(\frac{\mathbf{w} + \epsilon\mathbf{v}}{\|\mathbf{w} + \epsilon\mathbf{v}\|}\right) = \frac{1}{\|\mathbf{w} + \epsilon\mathbf{v}\|}\left(G(\mathbf{w}) - \epsilon\sum_{i=1}^n \ell_i'(0)\, \mathbb{1}\{(\mathbf{w} + \epsilon\mathbf{v})^\mathsf{T}\mathbf{x}_i > 0\}\, \mathbf{v}^\mathsf{T}\mathbf{x}_i\right). \tag{37}$$

*Proof.* Let $g$ be defined as in eq. (31); then

$$G\left(\frac{\mathbf{w} + \epsilon\mathbf{v}}{\|\mathbf{w} + \epsilon\mathbf{v}\|}\right) = \frac{1}{\|\mathbf{w} + \epsilon\mathbf{v}\|}(\mathbf{w} + \epsilon\mathbf{v})^\mathsf{T} g(\mathbf{w} + \epsilon\mathbf{v}). \tag{38}$$

We now analyse $\mathbf{w}^\mathsf{T} g(\mathbf{w} + \epsilon\mathbf{v})$ and $\epsilon\mathbf{v}^\mathsf{T} g(\mathbf{w} + \epsilon\mathbf{v})$ separately, starting with the former. Denote $I_i^\epsilon := \mathbb{1}\{(\mathbf{w} + \epsilon\mathbf{v})^\mathsf{T}\mathbf{x}_i > 0\}$. Then $g(\mathbf{w} + \epsilon\mathbf{v})$ can be written as

$$g(\mathbf{w}+\epsilon\mathbf{v}) = -\sum_{i=1}^n \ell_i'(0)\cdot I_i^\epsilon I_i^0\,\mathbf{P}_\mathbf{w}\mathbf{x}_i - \sum_{i=1}^n \ell_i'(0)\cdot I_i^\epsilon(1-I_i^0)\,\mathbf{P}_\mathbf{w}\mathbf{x}_i - \sum_{i=1}^n \ell_i'(0)\cdot I_i^\epsilon\,(\mathbf{P}_{\mathbf{w}+\epsilon\mathbf{v}}-\mathbf{P}_\mathbf{w})\mathbf{x}_i. \tag{39}$$

Define

$$\epsilon_{\max} := \frac{1}{2}\max_{\epsilon>0}\epsilon, \qquad \text{subject to:}\ \ \text{sign}\left\{(\mathbf{w}+\epsilon\mathbf{v})^\mathsf{T}\mathbf{x}_i\right\}\text{sign}\left\{\mathbf{w}^\mathsf{T}\mathbf{x}_i\right\} \geq 0\ \ \forall i. \tag{40}$$

For $\epsilon \in [0, \epsilon_{\max}]$, $I_i^0 = 1$ implies $I_i^\epsilon = 1$, so the first term in eq. (47) equals $g(\mathbf{w})$. Regarding the second term, $I_i^\epsilon(1 - I_i^0)$ is nonzero only if $I_i^\epsilon = 1, I_i^0 = 0$. For $\epsilon \in [0, \epsilon_{\max}]$, this can only happen if $(\mathbf{w} + \epsilon\mathbf{v})^\mathsf{T}\mathbf{x}_i > 0$ and $\mathbf{w}^\mathsf{T}\mathbf{x}_i = 0$. In this scenario however, $\mathbf{P}_\mathbf{w}\mathbf{x}_i = \mathbf{0}$, so the second term in eq. (47) is zero. Regarding the third term, as long as $\epsilon \in [0, \epsilon_{\max}]$, $(\mathbf{w} + \epsilon\mathbf{v})^\mathsf{T}\mathbf{x}_i = 0$ implies $\mathbf{w}^\mathsf{T}\mathbf{x}_i = 0$, so

$$\mathbf{w} \in \text{span}\left\{\mathbf{x}_j|\,\mathbf{w}^\mathsf{T}\mathbf{x}_j = 0\right\}^\perp \subseteq \text{span}\left\{\mathbf{x}_j|\,(\mathbf{w}+\epsilon\mathbf{v})^\mathsf{T}\mathbf{x}_j = 0\right\}^\perp \tag{41}$$

and $\mathbf{w}^\mathsf{T}(\mathbf{P}_{\mathbf{w}+\epsilon\mathbf{v}} - \mathbf{P}_\mathbf{w}) = \mathbf{w}^\mathsf{T} - \mathbf{w}^\mathsf{T} = \mathbf{0}^\mathsf{T}$. It follows that

$$\mathbf{w}^\mathsf{T} g(\mathbf{w} + \epsilon\mathbf{v}) = \mathbf{w}^\mathsf{T} g(\mathbf{w}) = G(\mathbf{w}). \tag{42}$$

Turning to $\epsilon\mathbf{v}^\mathsf{T} g(\mathbf{w} + \epsilon\mathbf{v})$, we have that

$$\epsilon\mathbf{v}^\mathsf{T} g(\mathbf{w} + \epsilon\mathbf{v}) = -\epsilon\mathbf{v}^\mathsf{T}\sum_{i=1}^n \ell_i'(0)\cdot\mathbb{1}\{(\mathbf{w}+\epsilon\mathbf{v})^\mathsf{T}\mathbf{x}_i > 0\}\,\mathbf{P}_{\mathbf{w}+\epsilon\mathbf{v}}\mathbf{x}_i, \tag{43}$$

where

$$\epsilon\mathbf{v}^\mathsf{T}\mathbf{P}_{\mathbf{w}+\epsilon\mathbf{v}} = (\mathbf{w}+\epsilon\mathbf{v})^\mathsf{T}\mathbf{P}_{\mathbf{w}+\epsilon\mathbf{v}} - \mathbf{w}^\mathsf{T}\mathbf{P}_{\mathbf{w}+\epsilon\mathbf{v}} = (\mathbf{w}+\epsilon\mathbf{v})^\mathsf{T} - \mathbf{w}^\mathsf{T} = \epsilon\mathbf{v}^\mathsf{T}. \tag{44}$$

Plugging eq. (42) and eq. (43) into eq. (38) yields the result. $\qquad\square$

## A.2 TRAINING DYNAMICS

In the following lemma, we prove a formula for the evolution of the parameters of a two-layer network trained by gradient flow (eq. (4)). The formula has appeared in Maennel et al. (2018) before (but without a proof).

**Lemma A.3.** *Assume that the training inputs with the zero vector $\{\mathbf{x}_i\}_i \cup \{\mathbf{0}\}$ are in general position.[4] Then a two-layer ReLU network trained by gradient flow on $(\mathbf{X}, \mathbf{y})$ satisfies for all $j \in [p]$ and almost all $t \geq 0$,*

$$\frac{\partial a_j}{\partial t} = -\sum_{i=1}^n \ell_i'(t)\cdot\rho(\mathbf{w}_j^\mathsf{T}\mathbf{x}_i), \tag{45}$$

$$\frac{\partial \mathbf{w}_j}{\partial t} = -\sum_{i=1}^n \ell_i'(t)\,\mathbb{1}\left\{\mathbf{w}_j^\mathsf{T}\mathbf{x}_i > 0\right\}a_j\mathbf{P}_{\mathbf{w}_j}\mathbf{x}_i. \tag{46}$$

*Proof.* Fix $\boldsymbol{\theta}$, and denote by $\boldsymbol{\Sigma}_{\boldsymbol{\theta}} \in \{-1, 0, 1\}^{p\times n}$ the activation matrix for $f_{\boldsymbol{\theta}}$, $\boldsymbol{\Sigma}_{\boldsymbol{\theta}}[j, i] \triangleq \text{sign}\,\mathbf{w}_j^\mathsf{T}\mathbf{x}_i$. Then for any sequence $\boldsymbol{\theta}_k \to \boldsymbol{\theta}$ such that $\{\nabla\ell(\boldsymbol{\theta}_k)\}$ exists and has a limit,

$$\lim_{k\to\infty}\boldsymbol{\Sigma}_{\boldsymbol{\theta}_k} \in \left\{\boldsymbol{\Sigma} \in \{\pm 1\}^{p\times n}\,\big|\,\boldsymbol{\Sigma}[j, i] = \boldsymbol{\Sigma}_{\boldsymbol{\theta}}[j, i]\ \text{if}\ \boldsymbol{\Sigma}_{\boldsymbol{\theta}}[j, i] \neq 0\right\}. \tag{47}$$

Conversely, for any $\boldsymbol{\Sigma}$ in the set above, there exists a sequence $\boldsymbol{\theta}_k \to \boldsymbol{\theta}$ such that $\lim_{k\to\infty}\boldsymbol{\Sigma}_{\boldsymbol{\theta}_k} = \boldsymbol{\Sigma}$. To see this, observe that each $\mathbf{w}_j$ can be approached separately. Let $\mathbf{A}$ be the matrix whose rows are formed by those $\mathbf{x}_i$ for which $\mathbf{w}_j^\mathsf{T}\mathbf{x}_i = 0$. Then by the general position of inputs, $\mathbf{A}$ is a wide full-rank matrix and $\mathbf{A}\mathbf{w}_j = \mathbf{0}$. It follows that for any $\boldsymbol{\epsilon}$, $\mathbf{A}\mathbf{w} = \boldsymbol{\epsilon}$ has a solution, which can be chosen convergent to $\mathbf{w}_j$ as $\boldsymbol{\epsilon} \to \mathbf{0}$.

---

[4]That is, no $k$ of these points lie on a $(k - 2)$-dimensional hyperplane, for all $k \geq 2$.

We deduce that

$$\partial \ell(\boldsymbol{\theta}(t)) = \mathrm{conv} \left\{ \mathbf{g}(\boldsymbol{\Sigma}) \,|\, \boldsymbol{\Sigma}[j,i] \in \{\pm 1\}, \ \boldsymbol{\Sigma}[j,i] = \boldsymbol{\Sigma}_{\boldsymbol{\theta}(t)}[j,i] \text{ if } \boldsymbol{\Sigma}_{\boldsymbol{\theta}(t)}[j,i] \neq 0 \right\}, \qquad (48)$$

where we define $\mathbf{g}(\boldsymbol{\Sigma})$ coordinate-wise as

$$
\begin{aligned}
\mathbf{g}(\boldsymbol{\Sigma})[a_j] &:= \sum_{i=1}^{n} \ell_i'(t) \, \mathbb{1}\{\boldsymbol{\Sigma}[j,i] = 1\} \, \mathbf{w}_j^\mathsf{T} \mathbf{x}_i, \\
\mathbf{g}(\boldsymbol{\Sigma})[\mathbf{w}_j] &:= \sum_{i=1}^{n} \ell_i'(t) \, \mathbb{1}\{\boldsymbol{\Sigma}[j,i] = 1\} \, a_j \mathbf{x}_i.
\end{aligned}
\qquad (49)
$$

Since the value of $g(\boldsymbol{\Sigma})[a_j]$ is independent of $\boldsymbol{\Sigma}$, this proves eq. (45).

The proof of eq. (46) is slightly more complicated, as we need to pin down a single member of $\partial \ell(\boldsymbol{\theta}(t))$. To do that, we recall a result by Davis et al. (2020), who show that for a large class of deep learning scenarios (which includes ours), the objective $\ell$ admits a chain rule, i.e.

$$\ell'(t) = \left\langle \partial \ell(\boldsymbol{\theta}(t)), \frac{\partial \boldsymbol{\theta}}{\partial t} \right\rangle \qquad \text{for almost all } t \geq 0, \qquad (50)$$

where the right-hand side above should be interpreted as the only element of the set $\{\mathbf{h}^\mathsf{T} \partial \boldsymbol{\theta} / \partial t \,|\, \mathbf{h} \in \partial \ell(\boldsymbol{\theta}(t))\}$. For $t$ such that both eq. (4) and eq. (50) hold,

$$0 = \left\langle \partial \ell(\boldsymbol{\theta}(t)) - \partial \ell(\boldsymbol{\theta}(t)), \frac{\partial \boldsymbol{\theta}}{\partial t} \right\rangle, \qquad (51)$$

implying that

$$\frac{\partial \boldsymbol{\theta}}{\partial t} \in \mathrm{span} \left\{ \partial \ell(\boldsymbol{\theta}(t)) - \partial \ell(\boldsymbol{\theta}(t)) \right\}^\perp. \qquad (52)$$

Suppose $\mathbf{h}_1, \mathbf{h}_2$ satisfy both eq. (4) and eq. (52) (taking the role of $\partial \boldsymbol{\theta} / \partial t$). Then

$$\mathbf{h}_1 - \mathbf{h}_2 \in (\partial \ell(\boldsymbol{\theta}(t)) - \partial \ell(\boldsymbol{\theta}(t))) \cap \mathrm{span} \left\{ \partial \ell(\boldsymbol{\theta}(t)) - \partial \ell(\boldsymbol{\theta}(t)) \right\}^\perp = \{\mathbf{0}\}. \qquad (53)$$

It follows that $\partial \boldsymbol{\theta} / \partial t$ is the unique member of both $\mathrm{span} \{ \partial \ell(\boldsymbol{\theta}(t)) - \partial \ell(\boldsymbol{\theta}(t) \}^\perp$ and $-\partial \ell(\boldsymbol{\theta}(t))$. By eqs. (48) and (49),

$$\mathrm{span} \left\{ \partial \ell(\boldsymbol{\theta}(t)) - \partial \ell(\boldsymbol{\theta}(t)) \right\} \supseteq \mathrm{span} \left\{ \boldsymbol{\xi}_{ij} \,|\, \mathbf{w}_j^\mathsf{T} \mathbf{x}_i = 0 \right\}, \qquad (54)$$

where $\boldsymbol{\xi}_{ij}[\mathbf{w}_j] = \mathbf{x}_i$ and all other elements of $\boldsymbol{\xi}_{ij}$ are zero. Since

$$\frac{\partial \boldsymbol{\theta}}{\partial t} \in \mathrm{span} \left\{ \partial \ell(\boldsymbol{\theta}(t)) - \partial \ell(\boldsymbol{\theta}(t)) \right\}^\perp \subseteq \mathrm{span} \left\{ \boldsymbol{\xi}_{ij} \,|\, \mathbf{w}_j^\mathsf{T} \mathbf{x}_i = 0 \right\}^\perp, \qquad (55)$$

we obtain that for all $(i,j)$ with $\mathbf{w}_j^\mathsf{T} \mathbf{x}_i = 0$, $\partial \mathbf{w}_j / \partial t \perp \mathbf{x}_i$. In other words,

$$\frac{\partial \mathbf{w}_j}{\partial t} = \mathbf{P}_{\mathbf{w}_j} \frac{\partial \mathbf{w}_j}{\partial t} \in \mathrm{conv}_{\boldsymbol{\Sigma}} \left\{ -\sum_{i=1}^{n} \ell_i'(t) \, \mathbb{1}\{\boldsymbol{\Sigma}[j,i] = 1\} \, a_j \mathbf{P}_{\mathbf{w}_j} \mathbf{x}_i \right\}, \qquad (56)$$

where the inclusion follows from $\partial \boldsymbol{\theta} / \partial t \in -\partial \ell(\boldsymbol{\theta}(t))$ and eqs. (48) and (49). Now observe that by definition, $\mathbf{P}_{\mathbf{w}_j} \mathbf{x}_i = \mathbf{0}$ for all $(i,j)$ with $\boldsymbol{\Sigma}_{\boldsymbol{\theta}}[j,i] = 0$, hence the set in eq. (56) is a singleton whose only element equals eq. (46). $\qquad \square$

The following lemma shows that a balanced two-layer network remains balanced and neurons keep their signs.

**Lemma A.4.** *If a two-layer neural network is balanced at initialisation and trained by gradient flow with a loss whose derivative is bounded, then for $t \geq 0$,*

$$a_j(t) = \mathrm{sign} \, a_j(0) \cdot \|\mathbf{w}_j(t)\|. \qquad (57)$$

*Proof.* By Lemma A.3, for almost all $t \geq 0$,

$$\frac{\partial \|\mathbf{w}_j\|^2}{\partial t} = -\sum_{i=1}^n \ell_i'(t) \, \mathbb{1}\{\mathbf{w}_j^\mathsf{T}\mathbf{x}_i > 0\} \, a_j \mathbf{w}_j^\mathsf{T}\mathbf{x}_i = \frac{\partial a_j^2}{\partial t}, \tag{58}$$

i.e. $a_j$ and $\mathbf{w}_j$ grow equally fast. Since $|a_j(0)| = \|\mathbf{w}_j(0)\|$ at initialisation, $|a_j(t)| = \|\mathbf{w}_j(t)\|$ throughout training.

Next denote by $B, V > 0$ some scalars such that $|\ell_i'(u)| \leq B$ for all $i \in [n]$ and $u \in \mathbb{R}$, and $\|\mathbf{x}_i\| \leq V$ for all $i \in [n]$. Then $|\partial a_j^2/\partial t| \leq nBa_j^2 V$, or equivalently $|\partial \log a_j^2/\partial t| \leq nBV$. It follows that $a_j^2(t)$ lies between $a_j^2(0)\exp(-nBVt)$ and $a_j^2(0)\exp(nBVt)$, and hence $a_j$ cannot cross zero in finite time, proving eq. (57). $\qquad \square$

# B    PROOFS OF MAIN RESULTS

**Lemma 2.** *In the setting of Theorem 1, there is exactly one positive extremal direction and exactly one negative extremal direction. The positive extremal sector $\boldsymbol{\sigma}^+$ is given by*

$$\sigma_j^+ = \begin{cases} 1, & \text{if } y_j = 1, \\ -1, & \text{if } y_j = -1 \text{ and } \mathbf{x}_j^\mathsf{T}\mathbf{x}_i < 0 \text{ for some } i \text{ with } y_i = 1, \\ 0, & \text{if } y_j = -1 \text{ and } \mathbf{x}_j^\mathsf{T}\mathbf{x}_i = 0 \text{ for all } i \text{ with } y_i = 1, \end{cases} \tag{13}$$

*and the negative extremal sector $\boldsymbol{\sigma}^-$ is given by*

$$\sigma_j^- = \begin{cases} 1, & \text{if } y_j = -1, \\ -1, & \text{if } y_j = 1 \text{ and } \mathbf{x}_j^\mathsf{T}\mathbf{x}_i < 0 \text{ for some } i \text{ with } y_i = -1, \\ 0, & \text{if } y_j = 1 \text{ and } \mathbf{x}_j^\mathsf{T}\mathbf{x}_i = 0 \text{ for all } i \text{ with } y_i = -1. \end{cases} \tag{14}$$

*Proof.* We will prove the positive case; the negative case follows by inverting all labels. Because $G$ is a continuous function on a compact domain, it has a maximum. At least one maximum must moreover be strict, or otherwise $G$ would have to be constant. This shows that a positive extremal direction exists; we now show there is no more than one such direction.

By Lemma A.1, there cannot be more than one extremal direction per sector; it therefore suffices to show that no sector except one, $\boldsymbol{\sigma}^+$, admits a positive extremal direction. We will show that if $\mathbf{w} \in \mathbb{S}^{d-1}$ lies in any sector other than $\boldsymbol{\sigma}^+$, then $\mathbf{w}$ is not positive extremal; in particular we show that $G(\mathbf{w})$ can be locally increased.

Let $\boldsymbol{\sigma} \neq \boldsymbol{\sigma}^+$ and let $\mathbf{w} \in \mathbb{S}_{\boldsymbol{\sigma}} \cap \mathbb{S}^{d-1}$. By Lemma A.2, for any $\mathbf{v} \in \mathbb{R}^d$ there exists $\epsilon_{\max} > 0$ such that for $\epsilon \in (0, \epsilon_{\max}]$,

$$G\left(\frac{\mathbf{w} + \epsilon \mathbf{v}}{\|\mathbf{w} + \epsilon \mathbf{v}\|}\right) = \frac{G(\mathbf{w}) + \epsilon \alpha}{\|\mathbf{w} + \epsilon \mathbf{v}\|}, \tag{59}$$

where

$$\alpha := -\sum_{i=1}^n \ell_i'(0) \, \mathbb{1}\{(\mathbf{w} + \epsilon \mathbf{v})^\mathsf{T}\mathbf{x}_i > 0\}\mathbf{v}^\mathsf{T}\mathbf{x}_i. \tag{60}$$

We now analyse the different possible realisations of $\boldsymbol{\sigma}$, and for each we find $\mathbf{v} \in \mathbb{R}^d$ such that $(G(\mathbf{w}) + \epsilon \alpha)/\|\mathbf{w} + \epsilon \mathbf{v}\| > G(\mathbf{w})$ for small $\epsilon$.

Suppose first that $\sigma_j = -1$ for some example with $y_j = 1$, or that $\sigma_j = 1$ for some example with $y_j = -1$. Then set $\mathbf{v} := y_j \mathbf{x}_j / \|\mathbf{x}_j\|$. By orthogonal separability, we have that $\alpha \geq 0$. Also, $\sigma_j \triangleq \operatorname{sign} \mathbf{w}^\mathsf{T}\mathbf{x}_j = -y_j$ implies $\mathbf{w}^\mathsf{T}\mathbf{v} < 0$, therefore $\|\mathbf{w} + \epsilon \mathbf{v}\| < \|\mathbf{w}\| = 1$ for $\epsilon$ small enough. It follows that $(G(\mathbf{w}) + \epsilon \alpha)/\|\mathbf{w} + \epsilon \mathbf{v}\| > G(\mathbf{w})$.

Next suppose that $\sigma_j = 0$ for some example with $y_j = 1$, and set $\mathbf{v} := \mathbf{x}_j / \|\mathbf{x}_j\|$. Then $\alpha > 0$, because each term in eq. (60) is non-negative by orthogonal separability, and the term corresponding to $i = j$ is strictly positive:

$$-\ell_j'(0) \, \mathbb{1}\{(\mathbf{w} + \epsilon \mathbf{v})^\mathsf{T}\mathbf{x}_j > 0\} \, \mathbf{v}^\mathsf{T}\mathbf{x}_j = -\ell_j'(0) \, \mathbb{1}\{\epsilon \|\mathbf{x}_j\| > 0\}\|\mathbf{x}_j\| > 0. \tag{61}$$

From $\sigma_j \triangleq \operatorname{sign} \mathbf{w}^\mathsf{T} \mathbf{x}_j = 0$ it further follows that $\|\mathbf{w} + \epsilon \mathbf{v}\| = \sqrt{1 + \epsilon^2}$. Hence, $(G(\mathbf{w}) + \epsilon \alpha)/\|\mathbf{w} + \epsilon \mathbf{v}\| = (G(\mathbf{w}) + \epsilon \alpha)/(1 + O(\epsilon^2))$, which strictly exceeds $G(\mathbf{w})$ for $\epsilon$ small enough.

We have thus shown that if $\boldsymbol{\sigma}$ is positive extremal, then necessarily $\sigma_i = 1$ for all examples with $y_i = 1$, and $\sigma_i \in \{0, -1\}$ for examples with $y_i = -1$. Suppose now that $\sigma_j = 0$ for an example with $y_j = -1$ that satisfies $\mathbf{x}_j^\mathsf{T} \mathbf{x}_k < 0$ for some $k$ with $y_k = 1$. Taking $\mathbf{v} := -\mathbf{x}_j/\|\mathbf{x}_j\|$ will make $\alpha$ strictly positive, as the term corresponding to $i = k$ in eq. (60) will be strictly positive (this term's indicator equals 1, as we know from the above that $\sigma_k = 1$). Like in the previous paragraph, $\|\mathbf{w} + \epsilon \mathbf{v}\| = 1 + O(\epsilon^2)$, which suffices to show $G(\mathbf{w})$ is locally submaximal.

Finally, let $\mathbf{x}_j$ be such that $y_j = -1$ and $\mathbf{x}_j^\mathsf{T} \mathbf{x}_i = 0$ for all $i$ with $y_i = 1$, and suppose that $\sigma_j = -1$. With $\mathbf{v} := \mathbf{P}_\mathbf{w} \mathbf{x}_j/\|\mathbf{P}_\mathbf{w} \mathbf{x}_j\|$ (we know that $\mathbf{P}_\mathbf{w} \mathbf{x}_j \neq \mathbf{0}$ because $\mathbf{w}^\mathsf{T} \mathbf{x}_j \neq 0$ by $\sigma_j = -1$), we have

$$\alpha = -\frac{1}{\|\mathbf{P}_\mathbf{w} \mathbf{x}_j\|} \sum_{i=1}^n \ell_i'(0) \mathbb{1}\{(\mathbf{w} + \epsilon \mathbf{v})^\mathsf{T} \mathbf{x}_i > 0\} \mathbf{x}_j^\mathsf{T} \mathbf{P}_\mathbf{w} \mathbf{x}_i. \tag{62}$$

We claim that each term in eq. (62) is zero: For terms with $\sigma_i = -1$, the indicator $\mathbb{1}\{(\mathbf{w} + \epsilon \mathbf{v})^\mathsf{T} \mathbf{x}_i > 0\}$ is zero. For terms with $\sigma_i = 0$, $\mathbf{P}_\mathbf{w} \mathbf{x}_i = \mathbf{0}$. (Also notice that such terms necessarily satisfy $y_i = -1$ and $\mathbf{x}_i^\mathsf{T} \mathbf{x}_k = 0$ for all $k$ with $y_k = 1$, which we will need shortly.) Lastly, for terms with $\sigma_i = 1$, we know $y_i = 1$, and hence $\mathbf{x}_i^\mathsf{T} \mathbf{x}_l = 0$ for all $l$ with $\sigma_l = 0$. In other words, $\mathbf{x}_i \perp \operatorname{span}\{\mathbf{x}_l | \mathbf{w}^\mathsf{T} \mathbf{x}_l = 0\}$, implying $\mathbf{P}_\mathbf{w} \mathbf{x}_i = \mathbf{x}_i$. In the context of eq. (62), we obtain $\mathbf{x}_j^\mathsf{T} \mathbf{P}_\mathbf{w} \mathbf{x}_i = \mathbf{x}_j^\mathsf{T} \mathbf{x}_i = 0$, concluding the proof that $\alpha = 0$. Since $\sigma_j = -1$, $\|\mathbf{w} + \epsilon \mathbf{v}\| < \|\mathbf{w}\| = 1$ for small enough $\epsilon$, and $(G(\mathbf{w}) + \epsilon \alpha)/\|\mathbf{w} + \epsilon \mathbf{v}\| > G(\mathbf{w})$. We have thus ruled out all sectors except $\boldsymbol{\sigma}^+$, proving that for orthogonally separable datasets there is a unique positive extremal sector. $\square$

**Lemma 3.** *Assume the setting of Theorem 1. If at time $T$ the neuron $(a_j, \mathbf{w}_j)$ satisfies $a_j(T) > 0$ and $\mathbf{w}_j(T) \in \mathcal{S}_{\boldsymbol{\sigma}}$, where $\boldsymbol{\sigma}$ is the positive extremal sector (eq. (13)), then for $t \geq T$, $\mathbf{w}_j(t) \in \mathcal{S}_{\boldsymbol{\sigma}}$. The same holds if $a_j(T) < 0$ and $\boldsymbol{\sigma}$ is the negative extremal sector (eq. (14)).*

*Proof.* We omit the neuron index, and only prove the positive case; the negative case is analogous. Denote $\boldsymbol{\sigma}(t) := \operatorname{sign}(\mathbf{X}^\mathsf{T} \mathbf{w}(t))$. We proceed by contradiction. Suppose there exists a time $T_1 > T$ such that $\boldsymbol{\sigma}(T_1) \neq \boldsymbol{\sigma}(T)$. Wlog, take $T_1$ such that $\boldsymbol{\sigma}(t)$ is constant on $(T, T_1)$ and denote this constant sector $\bar{\boldsymbol{\sigma}}$; by continuity $\bar{\sigma}_k = \sigma_k(T)$ if $\sigma_k(T) \neq 0$.

Now consider $\sigma_k(T) = 0$. By the gradient flow differential inclusion, for almost all $t \in (T, T_1)$,

$$\frac{\partial \mathbf{w}^\mathsf{T} \mathbf{x}_k}{\partial t} \in \operatorname*{conv}_{\boldsymbol{\sigma}'} \left\{ -\sum_{i=1}^n \ell_i'(t) \mathbb{1}\{\sigma_i' = 1\} a \mathbf{x}_i^\mathsf{T} \mathbf{x}_k \right\}, \tag{63}$$

where each $\boldsymbol{\sigma}'$ in the definition of the convex hull satisfies $\sigma_i' = \sigma_i(T)$ if $\sigma_i(T) \neq 0$, implying

$$\begin{aligned} \{i \mid \sigma_i' = 1\} &\subseteq \{i \mid \sigma_i(T) = 1\} \cup \{i \mid \sigma_i(T) = 0\} \\ &= \{i \mid y_i = 1\} \cup \{i \mid y_i = -1 \text{ and } \mathbf{x}_i^\mathsf{T} \mathbf{x}_j = 0 \text{ for all } j \text{ with } y_j = 1\}. \end{aligned} \tag{64}$$

Denote the two sets in the last expression $\mathcal{I}^+$ and $\mathcal{I}^0$, and consider the gradient corresponding to some $\boldsymbol{\sigma}'$ in eq. (63). The gradient terms corresponding to $i \in \mathcal{I}^+$ are zero (because $k \in \mathcal{I}^0$ and so $\mathbf{x}_i^\mathsf{T} \mathbf{x}_k = 0$) and the terms corresponding to $i \in \mathcal{I}^0$ (if there are any) are negative. The total gradient for $\boldsymbol{\sigma}'$ is therefore non-positive, which is preserved under taking convex hulls, and so we obtain $\frac{\partial}{\partial t} \mathbf{w}^\mathsf{T} \mathbf{x}_k \leq 0$. It follows that $\bar{\sigma}_k \neq 1$.

By Lemma A.3, for almost all $t \in (T, T_1)$ and any $k \in [n]$,

$$\frac{\partial \mathbf{w}^\mathsf{T} \mathbf{x}_k}{\partial t} = -\sum_{i=1}^n \ell_i'(t) \mathbb{1}\{\bar{\sigma}_i = 1\} a \mathbf{x}_i^\mathsf{T} \mathbf{P}_\mathbf{w} \mathbf{x}_k, \tag{65}$$

where $\mathbb{1}\{\bar{\sigma}_i = 1\} = \mathbb{1}\{y_i = 1\}$ as we have shown above. Observe that for $\mathbf{x}_i$ with $y_i = 1$, we have $\mathbf{P}_\mathbf{w} \mathbf{x}_i = \mathbf{x}_i$. This is because $\mathbf{P}_\mathbf{w}$ projects onto

$$\operatorname{span}\{\mathbf{x}_i \mid \bar{\sigma}_i = 0\}^\perp \supseteq \operatorname{span}\{\mathbf{x}_i \mid \sigma_i(T) = 0\}^\perp \tag{66}$$

and $\mathbf{x}_i$ lies in the right-hand side by the definition of positive extremal sector (eq. (13)). Therefore

$$\frac{\partial \mathbf{w}^\mathsf{T} \mathbf{x}_k}{\partial t} = -\sum_{i=1}^n \ell_i'(t) \mathbb{1}\{y_i = 1\} a \mathbf{x}_i^\mathsf{T} \mathbf{x}_k. \tag{67}$$

One can easily check that if $\sigma_k(T) = 1$ then $\frac{\partial}{\partial t}\mathbf{w}^\mathsf{T}\mathbf{x}_k > 0$, if $\sigma_k(T) = -1$ then $\frac{\partial}{\partial t}\mathbf{w}^\mathsf{T}\mathbf{x}_k < 0$, and if $\sigma_k(T) = 0$ then $\frac{\partial}{\partial t}\mathbf{w}^\mathsf{T}\mathbf{x}_k = 0$. It follows that $\boldsymbol{\sigma}(T_1) = \boldsymbol{\sigma}(T)$, which is a contradiction. $\qquad\square$

**Lemma B.1.** *Assume the setting of Theorem 1. If at time $T$ the neuron $(a_j, \mathbf{w}_j)$ satisfies $a_j(T) > 0$ and $\mathbf{w}_j(T)^\mathsf{T}\mathbf{x}_k > 0$ for some $k \in [n]$ with $y_k = 1$, then for $t \geq T$, $\mathbf{w}_j(t)^\mathsf{T}\mathbf{x}_k > 0$. The same holds if instead $a_j(T) < 0$ and $y_k = -1$.*

*Proof.* We omit the neuron index, and only prove the positive case; the negative case is analogous. We will show that for almost all $t \in (T, \infty)$, $\partial\mathbf{w}^\mathsf{T}\mathbf{x}_k/\partial t \geq 0$. By the gradient flow differential inclusion, for almost all $t \in (T, \infty)$,

$$\frac{\partial\mathbf{w}^\mathsf{T}\mathbf{x}_k}{\partial t} \in \operatorname*{conv}_{\boldsymbol{\sigma}'}\left\{-\sum_{i=1}^n \ell_i'(t)\,\mathbb{1}\{\sigma_i' = 1\}\,a\mathbf{x}_i^\mathsf{T}\mathbf{x}_k\right\}. \tag{68}$$

Fix any $\boldsymbol{\sigma}'$ and consider the summand corresponding to example $i$. If $y_i = 1$, then $-\ell_i'(t) > 0$ and $\mathbf{x}_i^\mathsf{T}\mathbf{x}_k > 0$, so the summand is non-negative. If $y_i = -1$, then $-\ell_i'(t) < 0$ and $\mathbf{x}_i^\mathsf{T}\mathbf{x}_k \leq 0$, so the summand is again non-negative. It follows that the sum is non-negative irrespective of $\boldsymbol{\sigma}'$, hence $\partial\mathbf{w}^\mathsf{T}\mathbf{x}_k/\partial t \geq 0$. $\qquad\square$

**Corollary 1.** *Under the conditions of Theorem 1, there exist constants $u, z \geq 0$ such that*

$$\frac{f_{\boldsymbol{\theta}(t)}(\mathbf{x})}{\|\boldsymbol{\theta}(t)\|^2} \to u\rho(\mathbf{w}_+^\mathsf{T}\mathbf{x}) - z\rho(\mathbf{w}_-^\mathsf{T}\mathbf{x}), \qquad\qquad \text{as } t \to \infty. \tag{10}$$

*Proof.* By Lemma A.4, $\|\boldsymbol{\theta}\|^2 = \|\mathbf{a}\|^2 + \|\mathbf{W}\|_F^2 = 2\|\mathbf{a}\|^2 = 2\|\mathbf{W}\|_F^2$. Then for any $\mathbf{x} \in \mathbb{R}^d$,

$$\frac{2f_{\boldsymbol{\theta}(t)}(\mathbf{x})}{\|\boldsymbol{\theta}(t)\|^2} = \frac{\mathbf{a}(t)^\mathsf{T}}{\|\mathbf{a}(t)\|}\,\rho\left(\frac{\mathbf{W}(t)\mathbf{x}}{\|\mathbf{W}(t)\|_F}\right). \tag{69}$$

Denote $\tilde{\mathbf{a}} := \lim_{t\to\infty}\mathbf{a}(t)/\|\mathbf{a}(t)\|$. Then by Theorem 1, as $t \to \infty$,

$$\frac{2f_{\boldsymbol{\theta}(t)}(\mathbf{x})}{\|\boldsymbol{\theta}(t)\|^2} \to \tilde{\mathbf{a}}^\mathsf{T}\rho\left(u\mathbf{w}_+^\mathsf{T}\mathbf{x} + z\mathbf{w}_-^\mathsf{T}\mathbf{x}\right) \tag{70}$$

$$= \tilde{\mathbf{a}}^\mathsf{T}\rho(u\mathbf{w}_+^\mathsf{T}\mathbf{x}) + \tilde{\mathbf{a}}^\mathsf{T}\rho(z\mathbf{w}_-^\mathsf{T}\mathbf{x}) \tag{71}$$

$$= \tilde{\mathbf{a}}^\mathsf{T}\mathbf{u}\,\rho(\mathbf{w}_+^\mathsf{T}\mathbf{x}) + \tilde{\mathbf{a}}^\mathsf{T}\mathbf{z}\,\rho(\mathbf{w}_-^\mathsf{T}\mathbf{x}), \tag{72}$$

where in eq. (71) we used the fact that for all $i \in [p]$, either $u_i = 0$ or $z_i = 0$, and in eq. (72) we used $\mathbf{u}, \mathbf{z} \geq \mathbf{0}$. Finally, since $\tilde{\mathbf{a}} = \mathbf{u}\|\mathbf{w}_+\| - \mathbf{z}\|\mathbf{w}_-\|$, we have

$$\tilde{\mathbf{a}}^\mathsf{T}\mathbf{u} = \|\mathbf{w}_+\|\mathbf{u}^\mathsf{T}\mathbf{u} \geq 0, \tag{73}$$

$$\tilde{\mathbf{a}}^\mathsf{T}\mathbf{z} = -\|\mathbf{w}_-\|\mathbf{z}^\mathsf{T}\mathbf{z} \leq 0, \tag{74}$$

which completes the proof. $\qquad\square$

## C  RELATIONSHIP TO NONLINEAR MAX-MARGIN

**Lemma C.1.** *Let $(\mathbf{X}, \mathbf{y})$ be an orthogonally separable dataset, let $\mathbf{w}_+, \mathbf{w}_-$ be defined as in eqs. (6) and (7) and let*

$$\widetilde{\mathbf{W}} := \mathbf{u}\mathbf{w}_+^\mathsf{T} + \mathbf{z}\mathbf{w}_-^\mathsf{T}, \tag{75}$$

$$\tilde{\mathbf{a}} := \|\mathbf{w}_+\|\mathbf{u} - \|\mathbf{w}_-\|\mathbf{z}, \tag{76}$$

*for some $\mathbf{u}, \mathbf{z} \in \mathbb{R}_+^p$ such that $u_i = 0$ or $z_i = 0$ for all $i \in [p]$. Also let $\mathbf{u}, \mathbf{z}$ be normalised such that $\|\mathbf{u}\| = \|\mathbf{w}_+\|^{-1/2}$ and $\|\mathbf{z}\| = \|\mathbf{w}_-\|^{-1/2}$. Then $\tilde{\boldsymbol{\theta}} \triangleq \{\widetilde{\mathbf{W}}, \tilde{\mathbf{a}}\}$ is a KKT point of the following constrained optimisation problem:*

$$\min \frac{1}{2}\|\boldsymbol{\theta}\|^2, \qquad s.t. \qquad y_i f_{\boldsymbol{\theta}}(\mathbf{x}_i) \geq 1, \quad i \in [n]. \tag{77}$$

*Proof.* By standard linear max-margin considerations (e.g. Hastie et al. (2008, Section 4.5.2)), we know that

$$\mathbf{w}_+ = \sum_{i:y_i=1} \alpha_i \mathbf{x}_i, \qquad\qquad \mathbf{w}_- = \sum_{i:y_i=-1} \alpha_i \mathbf{x}_i, \qquad (78)$$

for some $\alpha_i \geq 0$ such that $\alpha_i = 0$ if $\mathbf{x}_i$ is a non-support vector. It follows by orthogonal separability that for $i$ with $y_i = 1$,

$$\mathbf{w}_+^\mathsf{T} \mathbf{x}_i > 0, \qquad\qquad y_i \mathbf{w}_-^\mathsf{T} \mathbf{x}_i \leq 0, \qquad (79)$$

and for $i$ with $y_i = -1$,

$$\mathbf{w}_+^\mathsf{T} \mathbf{x}_i \leq 0, \qquad\qquad y_i \mathbf{w}_-^\mathsf{T} \mathbf{x}_i > 0; \qquad (80)$$

we will need these properties shortly.

Let us now turn to checking the KKT status of $\tilde{\boldsymbol{\theta}}$ wrt. eq. (77). We start by showing that $\tilde{\boldsymbol{\theta}}$ is feasible. Let $\mathbf{x}_i$ be such that $y_i = 1$; then

$$y_i f_{\tilde{\boldsymbol{\theta}}}(\mathbf{x}_i) = \tilde{\mathbf{a}}^\mathsf{T} \rho(\widetilde{\mathbf{W}} \mathbf{x}_i) \qquad (81)$$
$$= \tilde{\mathbf{a}}^\mathsf{T} \rho(\mathbf{u}\mathbf{w}_+^\mathsf{T} \mathbf{x}_i + \mathbf{z}\mathbf{w}_-^\mathsf{T} \mathbf{x}_i) \qquad (82)$$
$$= \tilde{\mathbf{a}}^\mathsf{T} \mathbf{u}\, \mathbf{w}_+^\mathsf{T} \mathbf{x}_i \qquad (83)$$
$$= \|\mathbf{w}_+\| \|\mathbf{u}\|^2 \mathbf{w}_+^\mathsf{T} \mathbf{x}_i \qquad (84)$$
$$= \mathbf{w}_+^\mathsf{T} \mathbf{x}_i \geq 1, \qquad (85)$$

where the last inequality follows from the definition of $\mathbf{w}_+$, eq. (6). Similarly, if $\mathbf{x}_i$ is such that $y_i = -1$, then

$$y_i f_{\tilde{\boldsymbol{\theta}}}(\mathbf{x}_i) = -\tilde{\mathbf{a}}^\mathsf{T} \mathbf{z}\, \mathbf{w}_-^\mathsf{T} \mathbf{x}_i \qquad (86)$$
$$= \|\mathbf{w}_-\| \|\mathbf{z}\|^2 \mathbf{w}_-^\mathsf{T} \mathbf{x}_i \qquad (87)$$
$$= \mathbf{w}_-^\mathsf{T} \mathbf{x}_i \geq 1. \qquad (88)$$

This shows that $\tilde{\boldsymbol{\theta}}$ is feasible.

Next, we show that $\tilde{\boldsymbol{\theta}}$ is a KKT point, i.e. we show that there exist $\lambda_1, \ldots, \lambda_n \geq 0$ such that

1. for all $i \in [n]$, $\lambda_i \big( y_i f_{\tilde{\boldsymbol{\theta}}}(\mathbf{x}_i) - 1 \big) = 0$, and

2. $\tilde{\boldsymbol{\theta}} \in \sum_{i=1}^n \lambda_i y_i \partial_{\boldsymbol{\theta}} f_{\tilde{\boldsymbol{\theta}}}(\mathbf{x}_i)$,

where $\partial_{\boldsymbol{\theta}} f_{\tilde{\boldsymbol{\theta}}}(\mathbf{x})$ denotes the Clarke subdifferential of $f_{\boldsymbol{\theta}}(\mathbf{x})$ wrt. $\boldsymbol{\theta}$, evaluated at $\tilde{\boldsymbol{\theta}}$. Specifically, we show that the choice

$$\lambda_i = \begin{cases} \alpha_i / \|\mathbf{w}_+\|, & \text{if } y_i = 1, \\ \alpha_i / \|\mathbf{w}_-\|, & \text{if } y_i = -1, \end{cases} \qquad (89)$$

satisfies both conditions above.

As for the first condition, observe that if $i$ is a non-support example, then $\lambda_i = \alpha_i = 0$ and the condition holds. If $i$ is a support example, then by eqs. (85) and (88), $y_i f_{\tilde{\boldsymbol{\theta}}}(\mathbf{x}_i) = 1$ and the condition holds as well.

As for the second condition, denote

$$g_i(\boldsymbol{\theta}) := [\mathbf{I}_{\boldsymbol{\theta}}(\mathbf{x}_i)\mathbf{a}\mathbf{x}_i^\mathsf{T}; \ \rho(\mathbf{W}\mathbf{x}_i)], \qquad (90)$$

where $\mathbf{I}_{\boldsymbol{\theta}}(\mathbf{x}) = \mathrm{diag}\left[\mathbb{1}\{\mathbf{W}\mathbf{x}_i > \mathbf{0}\}\right] \in \mathbb{R}^{p \times p}$ is the diagonal matrix whose $(i,i)$-element is one if $\mathbf{w}_i^{\mathsf{T}}\mathbf{x} > 0$ and zero otherwise. It holds that $g_i(\tilde{\boldsymbol{\theta}}) \in \partial_{\boldsymbol{\theta}} f_{\tilde{\boldsymbol{\theta}}}(\mathbf{x}_i)$ and

$$
\begin{aligned}
\sum_{i=1}^{n} \lambda_i y_i g_i(\tilde{\boldsymbol{\theta}}) &= \sum_{i:y_i=1} \frac{\alpha_i}{\|\mathbf{w}_+\|} \left[\mathrm{diag}\left[\mathbb{1}\{\mathbf{u} > \mathbf{0}\}\right] \tilde{\mathbf{a}}\mathbf{x}_i^{\mathsf{T}};\ \rho(\mathbf{u}\mathbf{w}_+^{\mathsf{T}}\mathbf{x}_i + \mathbf{z}\mathbf{w}_-^{\mathsf{T}}\mathbf{x}_i)\right] \\
&\quad - \sum_{i:y_i=-1} \frac{\alpha_i}{\|\mathbf{w}_-\|} \left[\mathrm{diag}\left[\mathbb{1}\{\mathbf{z} > \mathbf{0}\}\right] \tilde{\mathbf{a}}\mathbf{x}_i^{\mathsf{T}};\ \rho(\mathbf{u}\mathbf{w}_+^{\mathsf{T}}\mathbf{x}_i + \mathbf{z}\mathbf{w}_-^{\mathsf{T}}\mathbf{x}_i)\right] \\
&= \sum_{i:y_i=1} \frac{\alpha_i}{\|\mathbf{w}_+\|} \left[\|\mathbf{w}_+\|\mathbf{u}\mathbf{x}_i^{\mathsf{T}};\ \mathbf{u}\mathbf{w}_+^{\mathsf{T}}\mathbf{x}_i\right] - \sum_{i:y_i=-1} \frac{\alpha_i}{\|\mathbf{w}_-\|} \left[-\|\mathbf{w}_-\|\mathbf{z}\mathbf{x}_i^{\mathsf{T}};\ \mathbf{z}\mathbf{w}_-^{\mathsf{T}}\mathbf{x}_i\right] \\
&= \frac{1}{\|\mathbf{w}_+\|} \left[\|\mathbf{w}_+\|\mathbf{u}\mathbf{w}_+^{\mathsf{T}};\ \mathbf{u}\mathbf{w}_+^{\mathsf{T}}\mathbf{w}_+\right] - \frac{1}{\|\mathbf{w}_-\|} \left[-\|\mathbf{w}_-\|\mathbf{z}\mathbf{w}_-^{\mathsf{T}};\ \mathbf{z}\mathbf{w}_-^{\mathsf{T}}\mathbf{w}_-\right] \\
&= \left[\mathbf{u}\mathbf{w}_+^{\mathsf{T}} + \mathbf{z}\mathbf{w}_-^{\mathsf{T}};\ \mathbf{u}\|\mathbf{w}_+\| - \mathbf{z}\|\mathbf{w}_-\|\right] \\
&\triangleq \tilde{\boldsymbol{\theta}}.
\end{aligned}
$$

This proves the second condition, and shows that $\tilde{\boldsymbol{\theta}}$ is a KKT point of eq. (77). $\qquad\square$

## D  EXPERIMENTS ON REAL DATA

In this section we explore the applicability of our result to real-world datasets and architectures (which lie outside the scope formally covered by our assumptions). We experiment on the MNIST dataset subsetted to two classes, the digit `0` and the digit `1`.

We train a network consisting of six convolutional layers followed by two fully-connected layers. We view the six convolutional layers as a 'feature extractor' and the two fully-connected layers as a two-layer fully-connected network of the kind we analyse in this paper. The details of the architecture are given in Table 1. We train the network by Adam with the binary cross-entropy loss and a batch size of 128. We train for 50 epochs. Prior to training, we multiply the weights of the fully-connected layers by 0.05, to approximate the small-norm initialisation assumed by theory.

| Layer | Type |
|-------|------|
| 1 | conv(32, 3, 1, 1) |
| 2 | conv(32, 3, 1, 1) |
| 3 | conv(32, 5, 2, 2) |
| 4 | conv(64, 3, 1, 1) |
| 5 | conv(64, 3, 1, 1) |
| 6 | conv(64, 5, 2, 2) |
| 7 | fc(3136, 128) |
| 8 | fc(128, 1) |

Table 1: Architecture of the studied network. By `conv(n, k, s, p)` we denote a convolutional layer with $n$ kernels of size $k \times k$ with stride $s$, where the input to the layer is padded by $p$ rows or columns on each margin. By `fc(n, o)` we denote a fully-connected layer whose input dimension is $n$ and whose output dimension is $o$.

We conduct two sub-studies. First, we demonstrate that the network learns orthogonally separable representations all by itself, in the course of training. This is shown in Figure 3. The first subplot shows three distributions: The blue distribution is the distribution of $\mathbf{x}_i^{\mathsf{T}}\mathbf{x}_j$ where $\mathbf{x}_i$ is sampled from class `0` and $\mathbf{x}_j$ is sampled from class `1`. The orange (or green) distribution is the distribution of $\mathbf{x}_i^{\mathsf{T}}\mathbf{x}_j$ where both $\mathbf{x}_i, \mathbf{x}_j$ are sampled from class `0` (or class `1`). The other subplots show analogous inner-product distributions for the intermediate representations or learned features of the data, i.e. $f_{\boldsymbol{\theta}}^l(\mathbf{x}_i)^{\mathsf{T}} f_{\boldsymbol{\theta}}^l(\mathbf{x}_j)$ instead of $\mathbf{x}_i^{\mathsf{T}}\mathbf{x}_j$.

What we see is that the network learns representations such that examples of the same class are more similar to each other than examples of different classes – the orange and green distributions are generally more to the right compared to the blue distribution. Moreover, higher-layer representations

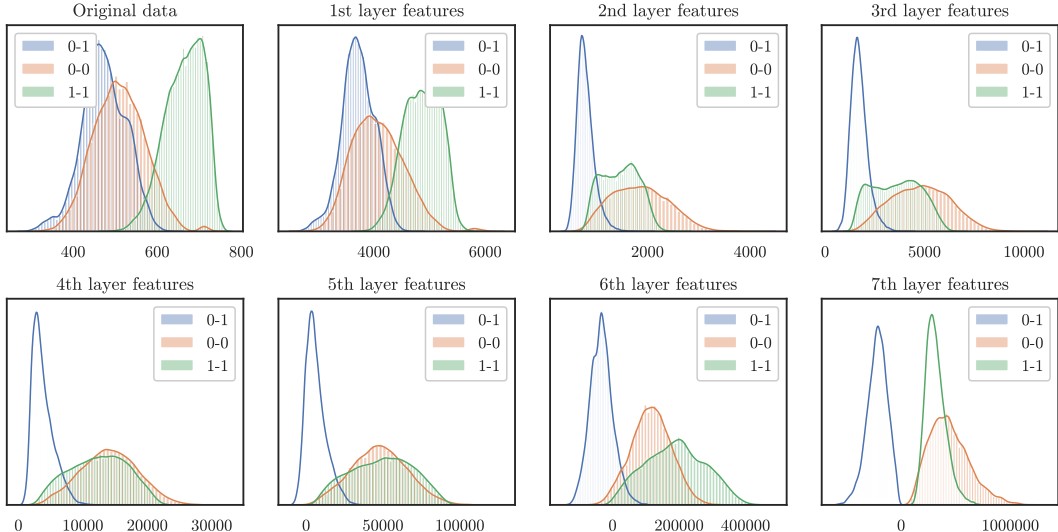

Figure 3: Distributions of feature similarity, where examples are sampled from the specified classes. Specifically, The $l$-th subplot shows the distribution of $f_{\theta}^{l}(\mathbf{x}_i)^{\intercal} f_{\theta}^{l}(\mathbf{x}_j)$ for $\mathbf{x}_i, \mathbf{x}_j$ sampled from different classes (blue) or both from class 0 (orange) or both from class 1 (green).

are generally more strongly separated – as we move up the layer hierarchy, the orange and green distributions keep shifting rightward, whereas the blue distribution shifts leftward. Remarkably, the 7th layer representations are orthogonally separable.

In the second sub-study, we explore properties of the weight matrix learnt by the first linear layer of the network, in analogy to the first-layer weight matrix in a two-layer net. Figure 4a shows the top ten singular values of the weight matrix $\mathbf{W}_7 \in \mathbb{R}^{128 \times 3136}$. We see that despite its size, it has very few (perhaps five or ten) significantly non-zero singular values. This is similar to what we observed for synthetic data in Section 5, though the separation between small and large singular values is less crisp and there are more than two non-zero values. Figure 4b shows the rows (neurons) of $\mathbf{W}_7$ projected onto the top two singular dimensions (note that unlike in Section 5, the projection is lossy). The neurons roughly form three clusters: a mixed cluster close to the origin and two clusters corresponding to positive and negative outer-layer weights. Compared to our observations from Section 5, there is less variation in the neurons' norms, leading to them forming clusters rather than rays. This deviation from the theoretical prediction could be due to a number reasons, e.g. the use of biases, convolutional layers, or the large dimensionality of the layer. We leave a detailed investigation of this question to future work.

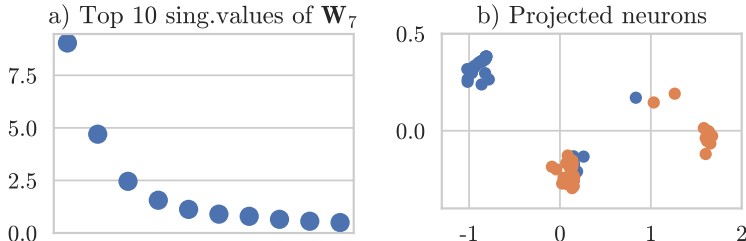

Figure 4: **a)** The 10 largest singular values of the first linear layer's weight matrix $\mathbf{W}_7$ after training. Each dot represents one singular value. **b)** Neurons (rows of $\mathbf{W}_7$) projected on the top two singular dimensions. Orange (or blue) dots represent neurons with $W_8[j] > 0$ (or $W_8[j] < 0$).

