# OpenReview forum: "The inductive bias of ReLU networks on orthogonally separable data"
_ICLR.cc/2021/Conference — ICLR 2021 Poster_

### Official Review · AnonReviewer1 · 2020-10-27
**I find that the paper is well written, modulo some assumptions on the data that would be better to be made more rigorous.**

**Rating:** 7
**Confidence:** 4

**Review:**

This paper studies the inductive bias of two-layer ReLU networks trained by gradient flow. The main challenge is to analyze the global convergence of the flow dynamics. Under a special assumption that the data are orthogonally separable, the paper shows that the dynamics converges to a unique max-margin solution. I find that the paper is well written, modulo some assumptions on the data that would be better to be made more rigorous. The overall quality is good. The novelty compared to the literature is that this paper provides a global analysis of the non-linear non-smooth dynamics without going into the over-parameterized regime.

In Theorem 1 of the paper, what does it mean « For almost all such datasets? » What is the probability distribution of (X,y)? Is there any more precision condition on \lambda, which controls the norm of the initial weights?

In Lemma A.3, what does it mean « genetic position »? I think what is needed is to assume that the probability that all the x_i lie on some hyperplane is close to zero. This assumption is crucial for (43) to hold, therefore I think it should be made more precise. Or to put in a less probabilistic, one may assume that the maximal number of samples {x_i} that lie on some hyperplane is smaller than the dimension of x_i.

For clarity, is the experiment in 5.2 uses the same data (X,y) as in 5.1?

Some typos or confusions of notations are listed below:
- \ell^+(t) right before (18), is confusing, as \ell^+ is a function of \theta, I would suggest to use L^+(t)  = \ell^+(\theta(t))
- Change W+(t) -> W_(t) to (24)
- \ell’_i(t) in (40) and (41) are also confusing, write L_i’(t) = \ell_i’(\theta(t)) ?
- Change \Sigma[i,j] -> \Sigma[j,i] in (42), (43), etc.

---

> ### Author Response · Authors · 2020-11-21
> **Thank you! Assumptions on the data explained in more detail and made formal**
>
> Thank you for your encouraging feedback!
> Below we hope to answer your questions, but please do comment if anything remains unclear.
>
> (General position) You are completely right. We mean 'general position' from geometry: no d-dimensional hyperplane contains more than d+1 points. We have now clarified this in the paper.
>
> (Almost all datasets) To motivate the framing, there are a number of 'bad configurations' that we would like to avoid. For example, Lemma 1 requires that for each sector, g(w) as defined in eq. (31) lies in the relative interior of the sector or in the interior of the complement of the sector (i.e. it does not lie exactly on the relative boundary of the sector). Another example is Lemma A.3 which, as you rightly point out, requires that the data lie in general position.
>
> One way to deal with all of these is to require that the data is sampled from a distribution with a density. More precisely, by 'almost all datasets' we mean that if x_i are sampled from any distribution with a density wrt the Lebesgue measure, then the theorem (treated as an implication) holds with probability one wrt the data. This is an easy-to-state and sufficient condition for our results to hold (we have now clarified this in the paper).
>
> (lambda) An explicit bound on lambda does seem attainable (see e.g. the last equation on p.26 of [Maennel et al, 2018]), but this goes deeper into prior work and we did not try to do it.
>
> (Experiments) Yes, the networks in 5.1 and 5.2 are trained on identical data.
>
> (Typos, notation) Thanks!

---

### Official Review · AnonReviewer4 · 2020-10-29
**Good paper that studies an important and interesting problem. Provides a clean solution.**

**Rating:** 8
**Confidence:** 4

**Review:**

This paper characterizes the implicit bias of gradient flow of two-layer ReLU networks on orthogonally separable data trained on the logistic loss. The problem of characterizing the implicit bias of gradient descent on neural networks is an important one, and while the authors do make fairly strong assumptions on the data (data corresponding to the different labels lie in separate orthants), the proof is novel, interesting and non-trivial. The proofs are carefully carried out and seemed as far as I could verify.


A few questions:
1. Is it possible to characterize what the outer weights (a's) converge to? If yes, I would suggest that the authors include this either in the main theorem, or as a comment after the theorem.
2. Does a similar result hold if the network also has bias variables?
3. Can linearly separable data be made into orthogonally separable data (by appropriate pre-processing) and by also training a bias term?
4. How are the lambda_j's chosen in the near zero initialization? The current description of choosing lambda_j on page 2 is quite vague.
5. I would also urge the authors to add the additional assumption about the positive and negative examples spanning the entire space in Section 2 along with the other assumptions.

---

> ### Author Response · Authors · 2020-11-21
> **Thank you!**
>
> Thank you for your encouraging feedback!
> Below we hope to answer your questions, but please do comment if anything remains unclear.
>
> 1, We have expanded Theorem 1 to include the characterisation of the outer weights as well. (The magnitude of a_j equals the norm of the respective w_j, and the sign of a_j is 1 if w_j converges to w+ and -1 if w_j converges to w-.)
>
> 2, First-layer biases: Yes, via the data transformation x' = [x, c] for a constant c. The constant would have to be small enough such that the transformed data is still orthogonally separable. The first-layer neurons would then converge to the bias-regularised max-margin directions. If the bias is also initialised differently, then this would affect the (1-1/2^p) factor, but the crux of the result should go through.
>
> Second-layer bias: Likely yes, but we are less sure. In this case, the (scalar) bias roughly tries to adjust so that the margin of the positive class equals the margin of the negative class. In the second phase of learning (after neurons have separated), this seems to matter little, but the first phase of training could be affected if the classes are very imbalanced.
>
> 3, The easiest (though not the most realistic) case is if the classes have disjoint sparsity patterns in feature space. Creating such class-specific features was popular e.g. for structured multiclass SVMs. In general, we are not aware of any simple preprocessing procedure that would guarantee orthogonal separability. However, in future work we'd be very interested in studying whether deep nets learn orthogonally separable representations by themselves (we have added some preliminary experiments in support of this in Appendix D).
>
> 4, There are no assumptions on lambda_j's except that they are small, lambda_j \in (0, lambda]. We have changed the text to make this a bit clearer.
>
> 5, Thank you, we have done so.

---

### Official Review · AnonReviewer2 · 2020-10-29
**The assumptions may be too strong**

**Rating:** 5
**Confidence:** 4

**Review:**

This paper studies the inductive bias of gradient flow for two-layer ReLU networks for classification problems. Under an orthogonally separable data assumption, it is shown that each node of the ReLU network will converge to one of two directions that linearly separate the data. I think the inductive bias of neural network training is a very important research problem and the result of this paper looks interesting. However, I also have the following concerns about this paper.

Perhaps the most obvious limitation of this paper is that the orthogonally separable data assumption is too strong. Under this assumption, the classification problem can be solved trivially: one can simply randomly pick a training example and use it as the parameters in a linear predictor. It seems to be highly unlikely that this assumption can be satisfied by any challenging real-world problems.

Moreover, the current submission lacks discussion and explanation of their results:

(a) The result of this paper seems to be weaker than the result in Lyu & Li (2020), while also requiring much stronger assumptions than Lyu & Li (2020).  Note that the inductive bias given in Lyu & Li (2020) is in the form of a maximum margin KKT point of the ReLU network (as a *nonlinear* classifier), however, the result in this paper is more related to the maximum margin solution of linear models, which in general may be much worse than the margin achievable by wide neural networks. Therefore I guess the most straightforward question the authors should clarify is whether under their setting w^+ and w^- indeed gives a KKT point of the *nonlinear* maximum margin problem given by Lyu & Li (2020).

(b) To my knowledge most of the inductive bias results for classification problems (cross-entropy/exponential loss) do not rely on specific initialization methods (except certain assumptions to guarantee achieving zero training error) (Soudry et al. (2018), Ji & Telgarsky (2019b), Lyu & Li (2020)). Therefore the authors may consider providing more explanation on why they require the specific initialization.

(c) In Section 6 it is mentioned that Li & Liang (2018) contain the training in the neighborhood of the (relatively large) initialization. While this is to some extent true, I find this comment not very convincing. When studying inductive bias, it is natural to restrict the training to a fixed training dataset, i.e. to treat the online SGD in Li & Liang (2018) as finite sum SGD by considering a uniform data distribution over training samples. In this case, since Li & Liang (2018) considers classification with cross-entropy loss, the weights will eventually go to infinity and therefore will not stay in the neighborhood of initialization forever, as is shown in Lyu & Li (2020). This is also true for other classification results in the lazy training setting including [1,2,3,4]. It seems that a combination of these results mentioned above and the result by Lyu & Li (2020), which has been discussed in Ji & Telgarsky, 2020 and [5], can already imply a much stronger result compared to this paper.




[1] Zou, Difan, Yuan Cao, Dongruo Zhou, and Quanquan Gu. "Stochastic Gradient Descent Optimizes Over-parameterized Deep ReLU Networks." arXiv preprint arXiv:1811.08888 (2018).

[2] Nitanda, Atsushi, Geoffrey Chinot, and Taiji Suzuki. "Gradient Descent can Learn Less Over-parameterized Two-layer Neural Networks on Classification Problems." arXiv preprint arXiv:1905.09870 (2019).

[3] Cao, Yuan, and Quanquan Gu. "Generalization bounds of stochastic gradient descent for wide and deep neural networks." NeurIPS 2019.

[4] Ji, Ziwei, and Matus Telgarsky. "Polylogarithmic width suffices for gradient descent to achieve arbitrarily small test error with shallow relu networks." ICLR 2020.

[5] Moroshko, Edward, Suriya Gunasekar, Blake Woodworth, Jason D. Lee, Nathan Srebro, and Daniel Soudry. "Implicit bias in deep linear classification: Initialization scale vs training accuracy." arXiv preprint arXiv:2007.06738 (2020).

---

> ### Author Response · Authors · 2020-11-21
> **The setting is simple, but has not been addressed in the literature**
>
> Thank you for your feedback!
> Below we hope to address your concerns, but please do comment if anything remains unclear or disputable.
>
> (Orthogonal separability) We agree that the assumption is strong and that it makes the classification problem trivial. However, the network doesn't know it's trivial and doesn't use the trivial algorithm.
> It is a priori unclear what a network would do even in this setting, and we believe it was valuable to work it out.
>
> (a. Comparison to Lyu & Li) We think the mentioned paper and ours are not directly comparable.
> First, that paper assumes that at some time T, the network converges to a zero-training-error solution, and focusses on how training unfolds from there. It does not discuss how one arrives at such a zero-training-error solution in the first place, or indeed which of the many attainable zero-training-error solutions are preferred by (S)GD.
> In contrast, we analyse the entire training process, from initialisation until convergence. The starting point for our analysis is random initialisation, as opposed to an assumed zero-error solution, and we characterise the solution preferred by GD.
>
> Second, our result is actually stronger, in the sense that we give an exact formula for the learned network, whereas they give a necessary first-order condition with potentially many solutions (in fact, which of the first-order points (S)GD picks likely depends on which zero-error basin is assumed as the starting point of the analysis).
>
> Of course, our (stronger) conclusions make use of stronger assumptions, and the work of Lyu & Li applies more generally.
>
> (a. Nonlinear max-margin) Yes, the solution described in Theorem 1 (with appropriately normalised u and z) is indeed a KKT point of the nonlinear max-margin problem (P) [Theorem 4.4, Lyu & Li]. We have included a proof of this claim in Appendix C.
>
> (b) Compared to the first two mentioned papers, the answer is the difference between ReLU neurons and linear neurons. ReLU updates are more local: when learning, linear neurons take into account all examples, whereas ReLU neurons take into account examples in their positive half-plane only. How a ReLU neuron updates and what trajectory it takes is therefore determined by its initial orientation.
>
> Compared to the last paper, the answer is related to point (a) above: we are interested in the early phase of training and bias towards specific minima, which the other paper simply assumes away.
>
> (c. Comparison to Li & Liang) It seems to us that training longer (i.e. letting weights go to infinity, and the training error to approach zero) in the setting of [Li & Liang, 2018] would require the network to keep increasing its width [Li & Liang, p.5, last line]: m \geq poly(1/epsilon), which however is the basis for setting the initialisation scale [p.3, assumption (A3)]: sigma = m^{-1/2}. So, to be able to train longer, one needs to start with a smaller init, which has the effect of limiting how far one can go.
>
> In contrast, our work allows the training time to go to infinity and the width of the network to be fixed.
>
> However, if we have somehow misunderunderstood your argument, please do correct us and we are happy to discuss.
>
> (c. Possibly stronger result) We do not (yet) see how the papers [Lyu & Li 2020, Ji & Telgarsky 2020, Moroshko et al 2020] imply a stronger result than ours. We are happy to discuss if you provide more details, e.g. which theorems/lemmas you have in mind.

---

> > ### Comment · AnonReviewer2 · 2020-11-25
> > **Re: possibly stronger results**
> >
> > I greatly appreciate your detailed response. I would like to clarify my comment on the ‘possibly stronger results’.
> >
> > As you mentioned in your response, Lyu & Li assumes that at some time T, the network converges to a zero-training-error solution. Based on this assumption, Lyu & Li shows the implicit bias of homogeneous models.
> >
> > On the other hand, by Li & Liang, Ji & Telgarsky or other recent results on training neural networks with logistic loss, as long as the network is wide enough, the network should be able to achieve zero training error at some point during the training. Since we only apply these results to show the guarantee for a zero-training-error solution, it is not necessary to require an infinitely wide network. (Under the assumption in the submission, the data are linearly separable, and therefore the width requirement can be polylog by Ji & Telgarsky.)
> >
> > Combining these results, for example, Ji & Telgarsky and Lyu & Li, we can see that as long as the data are non-parallel to each other, wide enough networks can reach a KKT point of the nonlinear maximum margin problem. In contrast, the submission shows the convergence to a linear maximum margin solution under a much stronger assumption.
> >
> > I hope the above clarifies (i) why I think the previous results may still be comparable to this paper, (ii) why it is not necessary to apply Li & Liang (or other similar results mentioned in the review with milder data assumptions) for an arbitrarily long training time or increasingly wide networks, and (iii) why I feel the combination of the various existing results may be stronger than the results in this paper (weaker assumptions, deeper networks, more general implicit bias).
> >
> > That being said, I agree that the results in this submission provide a more refined characterization of the implicit bias under the stronger assumption of orthogonal separability, and this setting has not been studied in the literature.

---

### Official Review · AnonReviewer3 · 2020-10-30
**Very special case of inductive bias, very well presented.**

**Rating:** 8
**Confidence:** 3

**Review:**

(a) This belongs to the literature of implicit bias/inductive bias, which has
gained a great deal of attention among theoretical enthusiasts with an
optimization leaning.

(b) The paper is carefully laid out and argued, and is at a nice level
of clarity and precision.

(c) The mathematical argumentation seems to me correct;
however I haven't checked line-by-line.

(d) The situation being studied is very very special
and doesn't much correspond to the big kahuna
deep learning. Nevertheless the intellectual clarity
of this special case is quite appealing.

(e) The implied conclusion seems rather special
as well. From one viewpoint it says that if you start
from the get-go with perfect
separation of a particular strong form,
then the future evolution of the training
can never spoil things. This is a very weak statement,
but I suppose if we can't get results here in a very special case
that we can understand well, then the
general situation is truly hopeless.

Specific Comments.

(1) Why is this max-margin if your constraints only consider one class. It seems to be more of a finding a minimum-norm vector aligned with all training data of that class. Its unclear why the concept of separating margins comes in.

(2) “Theory III: Dynamics and Generalization in Deep Networks” by Banburski et al. also considers general deep relu networks and shows that the resulting margins are max-margin—requiring only separability, not orthogonal separability. In addition, that paper uses traditional DE methods rather than relying on lesser-known extremal sector techniques. Can you discuss or highlight why the simpler example in this paper might lead to insights not found in the other paper.

(3) While the paper says that it is not directly applicable to deep nets, it draws motivation from the popularity of that literature. In that spirit, to justify such an evocation, can you show at least one experiment on a non-synthetic dataset such as MNIST/CIFAR/etc (perhaps even simplified with hand-engineered preprocessed features and subsetted to two-classes) that would support the potential connection to deep learning?

(4) Can you provide any evidence why datasets would become orthogonally separated? Is there some feature
engineering procedure that tends to produce orthogonal separation?

(5) In Figure 1, variance is strange: shows one big outlier, but the plotted projection shows two roughly-equal-magitude directions of variation.

(6) It is unclear how Definition 2 relates to strict extremal directions as defined by the sign patterns.

(7) G should be clarified: What G is and what it represents should be explained to make the results more insightful



 16m 50s

Type a message

---

> ### Author Response · Authors · 2020-11-21
> **Thank you!**
>
> Thank you for your encouraging feedback!
> Below we hope to answer your questions, but please do comment if anything remains unclear.
>
> (1) Our notion of max-margin is the single-class one originally employed e.g. by one-class support vector machines. It corresponds to the max-margin separator between the data and the origin, and it coincides with the minimum-norm vector subject to inner-product constraints.
>
> (2) The mentioned paper assumes that at some time T, the network converges to a zero-training-error solution, and focusses on how training unfolds from there. It does not discuss how one arrives at such a zero-training-error solution in the first place, or indeed which of the many attainable zero-training-error solutions are preferred by (S)GD. Also, the term 'max-margin' is used differently than in our work: they mean 'min_x f(x)', which depends on the network parameters and is a moving target.
>
> In contrast, we analyse the entire training process, from initialisation until convergence. The starting point for our analysis is random initialisation, as opposed to an assumed zero-error solution, and we characterise the solution preferred by GD. The characterisation of the solution is complete in the sense that it is an exact formula depending on the dataset only (as opposed to a stationarity-type necessary condition with potentially many solutions).
>
> (3) Thank you for the suggestion, we have added an MNIST experiment to Appendix D (p.18).
>
> (4) The easiest (though not the most realistic) case is if the classes have disjoint sparsity patterns in feature space. Creating such class-specific features was popular e.g. for structured multiclass SVMs. In general, we are not aware of any simple preprocessing procedure that would guarantee orthogonal separability. However, in future work we'd be very interested in studying whether deep nets learn orthogonally separable representations by themselves (see also Appendix D, Figure 3).
>
> (5) The first singular value corresponds to the horizontal direction in Figure 1b (the 'shared' direction of w+ and w-), and the second singular value corresponds to the vertical direction in Figure 1b (the direction in which w+, w- deviate from the shared direction).
>
> (6,7) Thanks for the feedback! We have added an intuitive explanation of how Definitions 1 and 2 relate to each other, and what the role of G is.

---

### Decision · Program_Chairs · 2021-01-07
**Final Decision**

**Decision:**

Accept (Poster)

**Comment:**

The paper shows that under a very restrictive assumption on the data, ReLU networks with one hidden layer and zero bias trained by gradient flow converge two a meaningful predictor provided that the network weights are randomly initialized with sufficiently small variances. While there is some overlap with a paper by Lyu & Li (2020), the paper under review establishes its results for networks with arbitrary widths whereas using the results of Lyu & Li (2020) works, at least so far, only for sufficiently wide networks. The assumption on the data is anything than realistic and actually any "simple, conventional" learning algorithm can easily learn in this regime. Nonetheless, getting meaningful results for neural networks is still a notoriously difficult task and for this reason, the paper deserves publication.